# WHEN LESS IS MORE: UNCOVERING THE ROBUSTNESS ADVANTAGE OF MODEL PRUNING

## ABSTRACT

The interplay between neural network pruning, a widely adopted approach for model compression, and adversarial robustness has garnered increasing attention. However, most existing work focuses on empirical findings, with limited theoretical grounding. In this paper, we address this gap by providing a theoretical analysis of how pruning influences adversarial robustness. We first show that the pruning strategy and associated parameters play a critical role in determining the robustness of the resulting pruned model. We then examine how these choices affect the optimality of pruning in terms of maintaining performance relative to the original model. Building on these results, we formalize the inherent trade-off between clean accuracy and adversarial robustness introduced by pruning, emphasizing the importance of balancing these competing objectives. Finally, we empirically validate our theoretical insights on different models and datasets, reinforcing our novel understanding of the adversarial implications of pruning. Our findings offer a principled foundation for designing pruning strategies that not only achieve model compression but also enhance robustness without additional constraints or cost, yielding a "free-lunch" benefit.

## 1 INTRODUCTION

Large-scale neural networks, typically based on the transformer architecture (Vaswani et al., 2017), have recently achieved remarkable success, driving advancements across a wide range of applications, particularly in generative modeling and representation learning. Characterized by billions of parameters and extensive training data requirements, these models have set state-of-the-art performance in diverse fields such as Computer Vision (CV) (Dosovitskiy et al., 2020; Liu et al., 2021), Natural Language Processing (NLP) (Touvron et al., 2023; Jiang et al., 2023; Devlin et al., 2019), and Time Series (TS) (Goswami et al., 2024; Liang et al., 2024). However, their considerable size leads to significant computational costs, which not only restrict their deployment in resource-constrained environments but also raise serious concerns regarding energy efficiency and scalability.

Given that deep learning models often operate in an over-parameterized regime, a substantial body of research (Han et al., 2016; Cheng et al., 2018; Dantas et al., 2024; Zhu et al., 2024) has focused on reducing model complexity while maintaining performance. Among the various techniques, model pruning, consisting of the removal of less important weights from a pre-trained model, has emerged as a promising approach. By encouraging sparsity in model parameters, pruning techniques aim to reduce model size with minimal accuracy loss. Strategies for pruning can be applied before (Lee et al., 2019; de Jorge et al., 2021), during (Evci et al., 2020), or after training (Benbaki et al., 2023; Sehwag et al., 2020); however, given the widespread reliance on pre-trained large models, post-training (or no re-training) pruning methods are particularly attractive in the current practical applications. Different approaches have been proposed to determine which parameters to prune, ranging from simple magnitude-based methods (Han et al., 2015) to more advanced data-driven and optimization-based strategies (Cheng et al., 2024), all seeking to balance sparsity and performance.

Parallel to the developments in model compression, another critical concern in deep learning is the vulnerability of neural networks to adversarial attacks (Goodfellow et al., 2015). Small, often imperceptible perturbations in input data can cause significant misclassifications, posing serious risks in safety-critical applications such as autonomous driving, finance, and healthcare. Extensive research has been conducted on both adversarial attack mechanisms (Tramer et al., 2020; Costa et al.,

2024; Biggio et al., 2013) and potential defense strategies (Madry et al., 2017; Akhtar et al., 2021; ENNADIR et al., 2024), yet the interplay between model compression (specifically pruning) and adversarial robustness remains an open research question. While some empirical studies suggest that pruning can either enhance or degrade robustness depending on the strategy employed (Jordao & Pedrini, 2021), a rigorous theoretical foundation explaining this phenomenon is still lacking.

In this work, we aim to bridge this gap by conducting a theoretical investigation into the relationship between pruning and adversarial robustness. Specifically, we examine how the choice of pruning parameters can influence the robustness and overall performance of the pruned model. To the best of our knowledge, we are the first to formally establish theoretical upper bounds that connect adversarial robustness and pruning performance, and consequently formalize the trade-off. We begin by introducing a framework definition of adversarial robustness in the context of model pruning. Building on this foundation, we explore how pruning choices affect a model's resilience to adversarial attacks and its predictive accuracy. By combining these insights, we characterize the trade-off between performance and adversarial robustness under model pruning. Our analysis culminates in formulating this trade-off by providing the corresponding upper-bounds to control it, through which the final user can identify optimal pruning strategies balancing robustness and accuracy. We validate our theoretical findings empirically on various models and datasets. The contributions can be summarized as follows:

- We formally define adversarial robustness in the context of pruned neural networks and establish theoretical upper bounds linking pruning performance to adversarial robustness.
- We conduct a theoretical analysis of how pruning strategy and associated parameters affect both robustness and model performance, and consequently characterize the trade-off between accuracy and adversarial resilience by providing the corresponding upper-bounds.
- We validate our theoretical insights through extensive experiments on various models and different adversarial attacks using benchmark datasets.

## 2 RELATED WORK

**Pruning.** Pruning techniques, as a key approach within the broader field of model compression, have been extensively studied in the literature (LeCun et al., 1989; Hagiwara, 1993; Luo et al., 2017; Han et al., 2015; He et al., 2017). The fundamental objective of pruning is to eliminate redundant or low-importance neural connections while preserving the model's predictive performance. Various criteria have been proposed to guide the pruning process. Among the most widely adopted approaches is *magnitude-based pruning*, which removes parameters with the smallest absolute values based on the assumption that they contribute least to the model output (Hagiwara, 1993; Han et al., 2015). Alternatively, *score-based pruning* methods select parameters according to their sensitivity or their estimated impact on the network's performance (Soltani et al., 2021; Lee et al., 2019). Beyond direct pruning strategies, techniques such as knowledge distillation (Hinton et al., 2015) and neural architecture search (Mushtaq et al., 2023) have been employed to construct smaller, more efficient sub-networks that approximate the performance of the original, larger models.

**Pruning and Adversarial Robustness.** In recent years, a growing body of research has investigated the relationship between model pruning and the adversarial robustness of deep neural networks. Notably, prior work (Jordao & Pedrini, 2021) has empirically demonstrated that pruning can serve as an implicit regularizer, mitigating overfitting to adversarial perturbations and thereby enhancing model robustness. Beyond observational studies, several approaches have proposed pruning strategies explicitly designed to improve adversarial robustness while achieving model compression. For instance, HYDRA (Sehwag et al., 2020) introduces a robustness-aware pruning framework by formulating the pruning process as an empirical risk minimization problem, employing stochastic gradient descent to optimize weight importance scores and selectively prune parameters that minimally impact adversarial robustness. Similarly, ANP-VS (Madaan et al., 2020) presents a pruning-based adversarial defense mechanism by integrating Bayesian pruning with a vulnerability suppression loss, aiming to remove highly distorted latent features that contribute to adversarial susceptibility. Finally, HARP (Zhao & Wressnegger, 2023) proposes a holistic pruning framework that jointly learns layer-wise compression rates and connection importance scores through an adversarially regularized min-max optimization, enabling non-uniform, aggressive pruning while preserving both natural accuracy and adversarial robustness.

Despite these empirical findings, the theoretical understanding of how pruning affects adversarial robustness remains limited. While existing studies (Guo et al., 2018; Jordao & Pedrini, 2021; Piras et al., 2025) offer valuable experimental evidence, they do not provide a formal explanation of why or how pruning influences robustness. Unlike prior works that propose pruning methods to enhance robustness, our focus is different: we aim to bridge this gap by developing a general theoretical framework that explains the relationship between pruning strategies and the inherent adversarial robustness of deep neural networks, providing therefore a strong basis to enhance this line of research.

## 3 PRELIMINARIES

In this section, we start by introducing some fundamental concepts that will be used afterwards in our work. Afterward, we formulate our problem setup, which will be considered in our analysis.

**Transformer-based Models.** Let $X \in \mathcal{X} \subseteq \mathbb{R}^{n \times d}$ denote a sequence of $n$ tokens, where each token $x_i \in \mathbb{R}^d$. The backbone of a transformer $h : \mathcal{X} \subseteq \mathbb{R}^{n \times d} \to \mathcal{Z} \subseteq \mathbb{R}^{n \times d}$, as introduced in (Vaswani et al., 2017), is the *self-attention* mechanism, which computes a weighted combination of all token representations. Specifically, given learnable *query*, *key*, and *value* parameter matrices $W^Q, W^K, W^V \in \mathbb{R}^{d \times (d/H)}$, the output of a single *attention head* AH for input $X$ is defined as:

$$\text{AH}(X) = \text{softmax}\left( \frac{(XW^Q)(XW^K)^\top}{\sqrt{d/H}} \right) (XW^V), \tag{1}$$

where $H$ denotes the number of parallel attention heads and $d/H$ is the dimension per head. In practice, multiple attention heads $\text{AH}_i$ are computed in parallel, then concatenated and projected using a learnable weight matrix $W^O \in \mathbb{R}^{d \times d}$, yielding the multi-head attention (MHA) operation:

$$\text{MH}(X) = \text{concat}\big(\text{AH}_1(X), \text{AH}_2(X), \ldots, \text{AH}_H(X)\big)W^O. \tag{2}$$

In addition, each Transformer block incorporates a residual connection, layer normalization (Lei Ba et al., 2016) and a position-wise feed-forward network (FFN).

**Multi-Layer Perceptron (MLP).** Let $X \in \mathcal{X} \subseteq \mathbb{R}^n$ (e.g., a flattened image). An MLP is a sequence of fully connected layers, where each layer applies an affine and non-linear transformation:

$$h^{(\ell)} = \sigma(W^{(\ell)} h^{(\ell-1)} + b^{(\ell)}), \text{ with } h^{(0)} = X.$$

**Pruning.** The central idea behind pruning is that over-parameterized models often contain many redundant or non-essential neuron connections, which can be removed without significantly affecting test accuracy. Given a weight matrix $W \in \mathbb{R}^{e \times d}$, the goal is to produce a pruned version $W' \in \mathbb{R}^{e \times d}$ with more zero entries. As discussed in Section 2, one widely used strategy is magnitude-based pruning, which removes weights with small magnitudes under the assumption that larger weights contribute more significantly to model predictions. Formally, this involves finding a mask $M = \text{Top}_p(S_{i,j}) \in \{0,1\}^{e \times d}$, where $S = \{\|W_{i,j}\| : 1 \le i \le e, 1 \le j \le d\}$, and $\text{Top}_p(\cdot)$ selects the top $p\%$ largest entries. Another family of approaches, known as score-based pruning, aims to remove weights that contribute the least to task-specific metrics, such as accuracy. Typically, this involves training (or fine-tuning) the model on a given task, computing an importance score for each weight based on its impact on the training objective, and then pruning accordingly. Concretely, a parallel score matrix $S \in \mathbb{R}^{e \times d}$ is learned during training to assess the importance of each weight, and a binary mask $M \in \{0,1\}^{e \times d}$ is applied based on these scores. In this work, we focus on these two families of pruning techniques. To model them in a unified way, we view pruning as a probabilistic mapping governed by Bernoulli random variables. Specifically, for each weight $W_{i,j}$, we define:

$$W'_{i,j} = \begin{cases} W_{i,j}, & \text{with probability } p_{i,j}, \\ 0, & \text{with probability } 1 - p_{i,j}. \end{cases}$$

**On the probabilistic Aspect.** By appropriately defining the probabilities $p_{i,j}$, we can represent different pruning strategies. For instance, in magnitude-based pruning that is based on using a threshold $q$, we set $p_{i,j} = 1$ if $\|W_{i,j}\| \ge q$, and $p_{i,j} = 0$ otherwise. This formulation can be extended to smoother variants by setting probabilities as a continuous function of the weight norm, such as $p_{i,j} = 1 - \exp(-\alpha \|W_{i,j}\|)$, where $\alpha$ is a smoothing parameter. In the case of score-based pruning, the

probabilities $p_{i,j}$ are directly derived from the learned scores $S_{i,j}$ during training. Consequently, the considered probabilistic representation of pruning is universal and provides a unified representation of pruning in its different forms and strategies as explored by previous work (Qian & Klabjan, 2021).

**Problem Setup.** Following the previous discussion, we consider a model $f$ that is either an MLP or a transformer-based model (TBM), where all activation functions are 1-Lipschitz continuous, an assumption that holds for most commonly used activation functions, such as ReLU and Tanh (Virmaux & Scaman, 2018). Without loss of generality, we focus on the space of images, specifically, we consider the model's input to be normalized, i.e., $\mathcal{X} \subseteq [0,1]^{n \times d}$.

## 4 PRUNING MEETS ADVERSARIAL ROBUSTNESS

In this section, we aim to theoretically analyze the link between adversarial robustness and model pruning. We start by introducing the concept of "vulnerability" of a model, and we afterwards provide an analysis in the case of Transformer-based Models and an MLP. In what follows, $\|\cdot\|$ denotes the spectral matrix norm (resp. Euclidean norm).

### 4.1 ADVERSARIAL ROBUSTNESS

Let us consider a trained classifier $f : \mathcal{X} \to \mathcal{Y}$ and let $x \in \mathcal{X}$ be an input with its associated label vectors $y \in \mathcal{Y}$, such that $f(x) = y$. The objective of an adversarial attack is to generate a perturbed version of the input $\tilde{x}$ which is slightly different from the original input $x$, and whose prediction is different from the original one. The adversarial aim can therefore be formulated as the search for a perturbed attributed graph $\tilde{x}$ within a defined similarity budget $\epsilon$, such that $f(\tilde{x}) \neq f(x)$. In this perspective, we start by defining the neighborhood of an input $x$ with respect to an attack budget $\epsilon$:

$$\mathcal{B}(x, \epsilon) = \{\tilde{x} \in \mathcal{X} : \|x - \tilde{x}\| \leq \epsilon\}$$

In addition, we assume that the model $f$ undergoes a pruning strategy $\tau_p(\cdot)$ with a set of parameters $p$, as introduced in Section 3, resulting in a pruned model denoted by $g$. Using the notion of input neighborhoods, we define the *expected adversarial risk* (adapted from Hein & Andriushchenko (2017); Rice et al. (2021)) of the pruned model $g$ as the expected output behavior of adjacent elements with the considered input's neighborhood within a budget $\epsilon$ under the pruning transformation.

$$\mathcal{R}_\epsilon[f, \tau_p] = \mathbb{E}_{g \sim \tau_p[f]} \left[ \mathbb{E}_{x \in \mathcal{D}_\mathcal{X}} \left[ \sup_{\tilde{x} \in \mathcal{B}(x, \epsilon)} d_\mathcal{Y}(g(\tilde{x}), g(x)) \right] \right], \tag{3}$$

with $d_\mathcal{Y}$ being any defined distances in the measurable output $\mathcal{Y}$. In the current analysis, we consider $\ell_2$-norm as our distance metric for both the input and output space. Note that there exists an equivalence in terms of norm, and therefore, this latter choice can easily be extended to other norms.

**Definition 1** (Adversarial Robustness). *The pruning strategy $\tau_p$ is said to be $(\epsilon, \gamma)$-robust if its adversarial risk with respect to the classifier $f$ satisfies: $\mathcal{R}_\epsilon[f, \tau_p] \leq \gamma$.*

In the adversarial setting, the objective is to ensure that the adversarial risk remains small, implying that model predictions are stable under small perturbations. In this perspective, Definition 1 provides the notion of adversarial robustness for a pruning strategy.

### 4.2 ON THE ROBUSTNESS OF PRUNED MODELS

We now theoretically examine the relationship between pruning and adversarial robustness. We begin by focusing on TBMs, specifically considering $f$ as a one-layer TBM model with $H$ self-attention heads, as described in Section 3. Lemma 1 characterizes the robustness properties of the original, non-pruned model $f$, corresponding to the special case where the pruning probability is set to zero.

**Lemma 1.** *Let $f : \mathcal{X} \to \mathcal{Y}$ be the original TBM-based classifier following the considered problem setup. We have that the pruning strategy $\tau_{p=1}$ (i.e., no pruning) is $(\epsilon, \gamma)$-robust, with:*

$$\gamma = \left(\frac{d}{d-1}\right)^2 C_1 C_2 \epsilon,$$

*with* $\quad C_1 = \left(1 + \|W_O\|\sqrt{H} \max_h \left[\|W^{V,h}\| \left[\frac{4}{\sqrt{d/H}} \|W^{Q,h}\| \|W^{K,h}\| + 1\right]\right]\right)$ *and* $C_2 = \left(1 + \|W_{FFN}\|\right)$

Next, we study the effect of applying a uniform pruning strategy, which is based on using the same pruning probability across different layers and connections. This leads to the following result linking the robustness of the pruned model $g$ to that of the original model $f$.

**Theorem 1.** *Let $f\colon \mathcal{X} \to \mathcal{Y}$ to our original TBM-based classifier following our problem setup. Let $g$ be its corresponding pruned version using a pruning strategy $\tau_p$, then $\tau_p$ is $(\epsilon, \gamma) - robust$ with:*

$$\gamma' \leq C\gamma,$$

$$with \quad C = \frac{1 + p^2\|W^O\|_F\sqrt{H}\alpha}{1 + \|W^O\|_F\sqrt{H}\alpha} \times \frac{1 + p\|W_{FFN}\|_F}{1 + \|W_{FFN}\|_F} \leq 1$$

$$and \quad \alpha = \max_h \left[ \|W^{V,h}\|_F \left( \frac{4}{\sqrt{d/H}}\|W^{Q,h}\|_F\|W^{K,h}\|_F + 1 \right) \right].$$

Theorem 1 establishes formally a link between adversarial robustness and model pruning. Specifically, we see that the adversarial risk of the pruned model $g$ is always smaller than its corresponding original model $f$. We can therefore conclude that, from a theoretical standpoint, pruning inherently preserves or enhances adversarial robustness compared to the non-pruned counterpart. The link between these two elements is illustrated using the constant $C \leq 1$, which depends on the weight norms of the weight matrices linked to the attention framework, the weight of the concatenation of heads and the corresponding FFN. While the above analysis specifically targets Transformer-based architectures, the underlying principles extend to other neural network models, such as convolutional neural networks (CNNs) and multilayer perceptrons (MLPs) as we will discuss in the next section.

**Remark.** Both the non-pruned and pruned models are analyzed under the same Lipschitz-based approximation, so comparing their robustness bounds is meaningful despite the true values being unknown. This parallels statistical learning theory, where a smaller generalization bound is taken as a principled indication of better performance under identical assumptions. By the same reasoning, a smaller robustness bound for the pruned model provides theoretical justification for greater robustness, which is further confirmed by our empirical validation.

**On the generalization to Multi-Layers TBMs.** We note that the current theoretical analysis focuses on a single-layer Transformer-based model; nonetheless, the results naturally extend to the multi-layer case. Specifically, a Transformer model with $L$ layers, denoted as $f^{(L)}$, can be expressed as a composition of $L$ single-layer functions: $f^{(L)}(x) = f^{(L-1)} \circ f^{(L-2)} \circ \cdots \circ f^{(1)}(x)$. Under this formulation, and following standard results from Lipschitz continuity, the overall adversarial risk bound $\gamma$ becomes a multiplicative composition of the bounds for each individual layer. As a result, our robustness framework remains applicable in deeper architectures (as validated in the experiments). Moreover, due to the multiplicative nature of the resulting bound, we expect the robustness effect of pruning to increase as the depth of the model increases, as observed empirically in Section 6.

## 5 ON THE CHOICE OF PRUNING PARAMETERS

### 5.1 PRUNING CHOICES AFFECT ADVERSARIAL ROBUSTNESS

In the previous section, we established a general connection between model pruning and adversarial robustness in the case of Transformer-based models, showing that pruning can, under certain conditions, enhance robustness. However, the earlier results focused on the transformer-based model and specifically the special case where $p$ is constant for all weights and neurons. In this section, we seek to understand how varying the pruning probabilities affects the resulting robustness, providing deeper insight into how pruning strategies interact with adversarial robustness, and accordingly could be optimized to enhance a model's resilience to these perturbations. We consider an $L$-layer MLP model, and we consider the general probabilistic pruning model introduced in Section 3, which captures both magnitude-based and score-based pruning schemes. In this setting, we assume full control over the pruning probabilities $p_{i,j}^{(\ell)}$ at each layer $\ell$. Understanding how these choices influence robustness is crucial: for magnitude pruning, it allows direct parameter selection; for score-based pruning, it suggests ways to design or regularize the score-learning objectives such as to promote robustness alongside performance.

**Theorem 2.** *Let $f : \mathcal{X} \to \mathcal{Y}$ be an L-Layers MLP classifier. Let $g$ be its corresponding pruned version obtained through our considered pruning strategy with probabilities $p_{i,j}^{(\ell)}$. Then the pruning strategy $\tau$ is $(\epsilon, \gamma)$-robust with:*

$$\gamma = P_L \prod_{\ell=1}^{L} \|W_f^{(\ell)}\|_F, \text{ with } \quad P_L = \prod_{\ell=1}^{L} \sqrt{\max_{i,j} p_{ij}^{(\ell)}}$$

Theorem 2 provides an explicit upper bound on the adversarial risk $\gamma$ for a pruned MLP model, as a function of the pruning strategy $\tau$. The bound incorporates two key components: the product of the Frobenius norms of the original model's weight matrices, and a pruning-dependent term $P_L$, which scales the risk according to the maximum pruning probabilities in each layer. Notably, the bound reduces to the adversarial risk of the original, non-pruned model when all probabilities are set to one (i. e., no pruning). This formulation highlights how pruning directly influences robustness: reducing the number of active weights leads to a smaller $P_L$, and consequently, a lower adversarial risk.

The insight here parallels our earlier observations in the context of transformers and pooling, where we have seen that pruning acts as a form of structural regularization that reduces the model's sensitivity to input perturbations. In particular, the theorem reveals that pruning individual layers contributes multiplicatively to the overall robustness. As the pruning probabilities approach zero (i.e., most weights are removed), the bound tends to zero, corresponding to a degenerate model with no predictive capacity, consistent with the intuitive behavior of an empty network. This result underscores the hidden effect of pruning not only for compression but also for improving adversarial robustness in MLPs and other architectures.

## 5.2 LINKING PRUNING TO PERFORMANCE

In the previous section, we analyzed how pruning strategies and corresponding parameters influence adversarial robustness. Specifically, Theorem 2 showed that maximum robustness, corresponding to a vanishing upper bound, is theoretically achieved as $p \to 0$. However, although aggressive pruning improves robustness, it can severely degrade the model's ability to preserve the original predictive information. In practice, since we do not know a priori if a given input $x_0 \in \mathcal{X}$ has been adversarially perturbed, it is essential to maintain a balance: ensuring the model remains robust while still preserving high accuracy on clean, non-attacked data. Thus, the pruning parameters must be carefully chosen to avoid sacrificing standard performance for the aim of better robustness. In this subsection, we aim to formalize and study this trade-off. To this end, we start by introducing the notion of $\zeta$-optimality for a considered pruning strategy.

**Definition 2** (Optimal Pruning). *Let $f$ be a classifier and $g$ its pruned version obtained via a pruning strategy $\tau_p$. The pruning strategy is said to be $\zeta$-optimal over the input set $\mathcal{X}$ if:*

$$\mathbb{E}_{\tau_p}[\|f(x) - g(x)\|] \leq \zeta.$$

Ideally, we would like the pruned model $g$ to produce outputs close to those of the original model $f$, ensuring similar classification performance. Definition 2 captures this objective since the smaller the value of $\zeta$, the more faithful and closer the pruned model is to the original. Naturally, the quantity $\zeta$ depends on the specific choice of pruning probabilities; we therefore study the optimality of our considered probabilistic pruning in the case of an $L$-layer MLP under the same previous setup.

**Proposition 1.** *Let $f$ be an L-layer MLP and $g$ be its corresponding pruned version obtained through our considered pruning strategy with probabilities $p_{i,j}^{(\ell)}$. For an input point $x_0 \in \mathcal{X}$, the chosen pruning strategy is $\zeta$-optimal, with:*

$$\zeta = \prod_{\ell=1}^{L} \|W^{(\ell)}\| \sum_{\ell=1}^{L} \frac{1}{\|W^{(\ell)}\|} \sqrt{\sum_{i,j} (1 - p_{i,j}^{(\ell)})(W_{i,j}^{(\ell)})^2}.$$

The result presented in Proposition 1 offers several important insights into how pruning parameters influence the performance and optimality of the pruned model. First, the bound $\zeta$ scales with the network depth $L$, indicating that deeper architectures amplify the effects of pruning errors. Moreover, the bound depends on the pruning probabilities weighted by the magnitudes of the corresponding

weights, reinforcing the intuitive idea that pruning weights with larger magnitudes leads to greater performance degradation. Additionally, layers with smaller weight norms $\|W^{(\ell)}\|$ are more vulnerable to pruning errors, as reflected by the $1/\|W^{(\ell)}\|$ term appearing in the summation. It is also noteworthy that $\zeta \to 0$ as $p \to 1$, meaning that as pruning vanishes (no weights are pruned), the pruned model perfectly recovers the original model's outputs. This latter observation highlights the existence of an inverse relationship between adversarial robustness and pruning optimality, formally demonstrating the existence of a trade-off between robustness and optimality performance.

## 5.3 On the Trade-off between Robustness and Optimality

In the previous section, we established two key results: (1) pruning improves adversarial robustness by reducing the upper bound $\gamma$, and (2) it impacts model optimality, as reflected in the $\zeta$-optimality criterion. Specifically, more aggressive pruning (i.e., smaller $p_{i,j}^{(\ell)}$) increases sparsity, leading to lower adversarial risk but potentially reducing the model's capacity to approximate the target function, resulting in a potential drop in accuracy.

Both robustness and optimality are explicit functions of the pruning probabilities $p_{i,j}^{(\ell)}$, revealing an inherent trade-off between the two. Consequently, when pruning to reduce computational or storage costs, users should carefully balance this trade-off to gain robustness without compromising performance. Our main finding is that with appropriate choices of pruning parameters, it is possible to improve robustness without additional constraints or overhead, offering a "free-lunch" gain.

This gain can be found through an optimization problem in which we try to minimize the computed upper-bounds, resulting in better adversarial robustness and keeping a satisfactory pruning performance. We formalize this trade-off as a bi-objective optimization $(a)$ for $p \in [0,1]$:

$$(a) \quad \min_p \quad \left( \zeta(p_{i,j}^{(\ell)}),\ \gamma(p_{i,j}^{(\ell)}) \right) \quad \Rightarrow \quad (b) \quad \min_p \quad \lambda\gamma(p_{i,j}^{(\ell)}) + (1-\lambda)$$

where formulation $(b)$ is a practical adaptation consisting of adopting a linear scalarization (Boyd & Vandenberghe, 2004) to combine the two objectives using a weighting parameter $\lambda \in [0,1]$ to govern the trade-off, with higher values favoring robustness and lower values prioritizing accuracy.

**Practical implementation.** Different techniques can be adopted to solve the problem; in our case, we adopt a fixed-point coordinate descent strategy resulting in Algorithm 1. Full derivation and algorithmic details are provided in Appendix E, along with experimental validation in Section 6.5.

**On the existence of a sweet spot.** Prior work (Guo et al., 2018; Jordao & Pedrini, 2021; Piras et al., 2025) has empirically shown a trade-off between robustness and performance. Using our derived upper bounds and optimization formulation, we can further theoretically characterize this trade-off and identify a Pareto front, revealing a natural sweet spot. Intuitively, the bounds show that robustness depends on $p$, while optimality depends on $1 - p$, highlighting their inverse relationship. More formally, we provide theoretical characterization in Appendix: Lemma 3 analyzes uniform pruning, while Remark 1 treats the general layer-wise case.

## 6 Experimental Validation

This section provides empirical validation of our theoretical findings by analyzing the effect of pruning parameters on both adversarial robustness and clean accuracy. We first describe the experimental setup, then present an empirical analysis of the adversarial risk, followed by an evaluation of how these findings translate into clean and attacked accuracy across different pruning configurations.

### 6.1 Experimental Setup

**Architecture.** While our theoretical analysis focused on multilayer perceptrons (MLPs), we seek to empirically assess the generalization of our conclusions across different model architectures. To this end, we consider three commonly used architectures in vision tasks: **(i)** a two-layer MLP, **(ii)** a Vision Transformer (ViT) (Vaswani et al., 2017), and **(iii)** a convolutional neural network (CNN).

**Datasets.** We conduct experiments on a diverse set of vision-based classification benchmarks, including MNIST, CIFAR-10, CIFAR-100 (Krizhevsky et al., 2009), and ImageNet-100 (Russakovsky

et al., 2015). Due to the limited capacity of MLPs on more complex datasets, we evaluate MLPs only on MNIST and CIFAR-10. For each model, we adapted the number of epochs to ensure convergence towards a satisfactory clean accuracy. Additional implementation details, including hyperparameters, are provided in Appendix I. The necessary code to reproduce our experiments is included in the supplementary materials and will be made publicly available upon publication.

**Attacks.** In addition to validating our theoretical results using the adversarial risk quantities $\gamma$ and $\zeta$, we evaluate robustness under different widely used adversarial attacks for image-based models. We consider Fast Gradient Sign Method (FGSM), Projected Gradient Descent (PGD), CW, and AutoAttack attacks, with extended results in Appendix H and implementation details in Appendix I.

## 6.2 EMPIRICAL ANALYSIS OF THE ROBUSTNESS AND OPTIMALITY

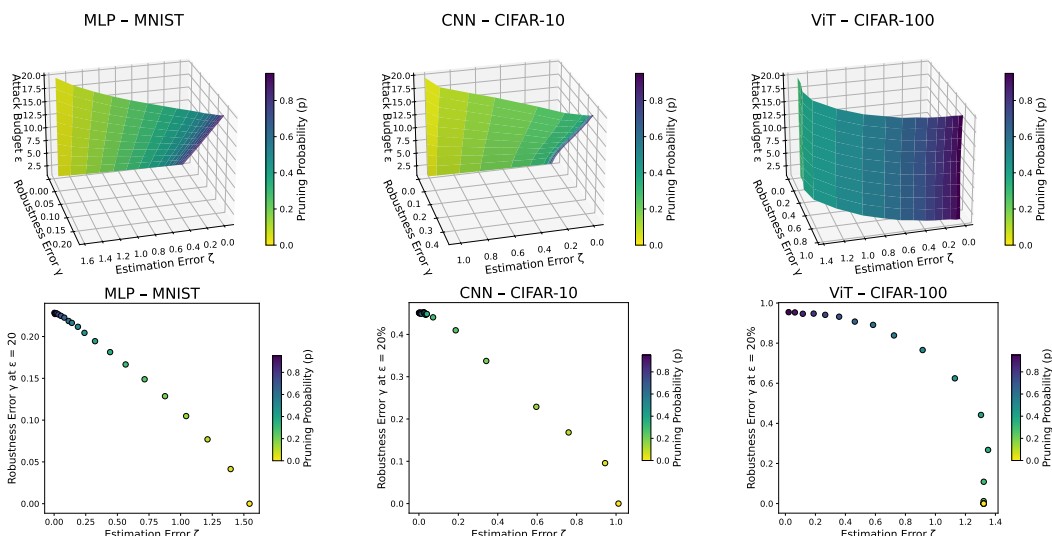

Figure 1: Empirical Analysis of the effect of the pruning parameters on the adversarial risk (Definition 1) and the Estimation Error (Equation 2) when subject to a range of attack budget ($\epsilon$).

We evaluate two key quantities introduced in our theoretical framework. First, we examine the adversarial risk, denoted by $\gamma$ in Definition 1 and upper-bounded in Theorem 2. Second, we assess the optimality of the pruning method, captured by the quantity $\zeta$ in Definition 2 and further analyzed in Proposition 1. Figure 1 illustrates the behavior of these quantities as functions of the pruning probability $p$, across different neighborhood sizes defined by $\epsilon$. We estimate adversarial risk by sampling $K$ points per $\epsilon$-neighborhood and computing average output divergence; for large $K$, this provides an unbiased estimator of Equation 3. We observe that experimental results closely align with the theoretical insights. Specifically, when the pruning probability approaches one (i. e., $p \to 1$), corresponding to minimal or no pruning, the pruning optimality metric satisfies $\zeta = 0$, and the adversarial risk $\gamma$ reflects the robustness of the original, non-pruned model. As we gradually decrease $p$ and increase the degree of pruning, we observe two simultaneous trends: the adversarial risk $\gamma$ decreases, indicating an increasing robustness, while the optimality $\zeta$ degrades, reflecting a growing deviation from the original model's behavior.

## 6.3 ON THE CLEAN/ATTACKED ACCURACY TRADE-OFF

**Magnitude-based pruning.** In the previous section, we studied the effect of pruning parameters on the adversarial risk and the optimality quantities. While this has already shown the existence of the studied trade-off, we are also interested in seeing how this trade-off translates into clean and attacked accuracy using real adversarial attacks. In this perspective, we consider the FGSM attack (while the PGD attack is reported in Appendix H), and study the effect of the pruning probability $p$ on the resulting clean accuracy of the pruned model (representing the optimality) and the attacked accuracy (representing the adversarial vulnerability) when subject to the considered attacks.

Figure 2 illustrates the trade-off between clean and adversarial accuracy across different pruning ratios for various model and dataset combinations. We observe a clear trend: as the pruning probability

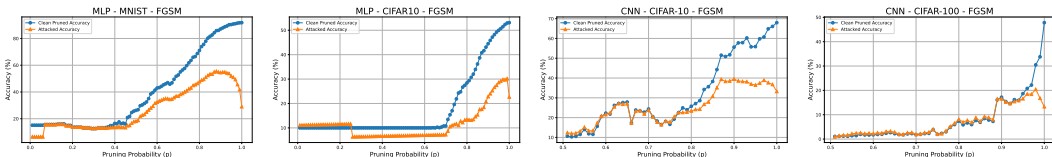

Figure 2: Clean and attacked accuracy of pruned models when subject to the FGSM attack across varying pruning probabilities and different datasets.

decreases from 1, adversarial accuracy initially improves, indicating enhanced robustness, while clean accuracy begins to decline. Beyond a certain point, further pruning causes a drop in both clean and adversarial accuracy, as the loss in representational capacity outweighs the robustness gains. These results highlight the presence of a retention "sweet spot" (for instance around $p = 0.9$ for MLP), where moderate pruning achieves an effective balance between robustness and clean performance.

**Score-based pruning.** To further demonstrate the generality of our theoretical insights, originally derived under probabilistic pruning, we extend the analysis to score-based pruning, where pruning decisions rely on gradient-based importance scores. In this setting, we apply pruning to a ViT model and evaluate it using our standard experimental setup. Figure 3 (and Figure 10 in Appendix H) shows the clean and attacked accuracies under FGSM (respectively PGD) for varying pruning ratios. Consistent with our earlier findings on magnitude-based pruning, we observe that pruning enhances adversarial robustness while introducing a trade-off with accuracy. Interestingly, in the score-based case, the trade-off occurs at higher pruning ratios, which we attribute to the greater precision of gradient-based scoring.

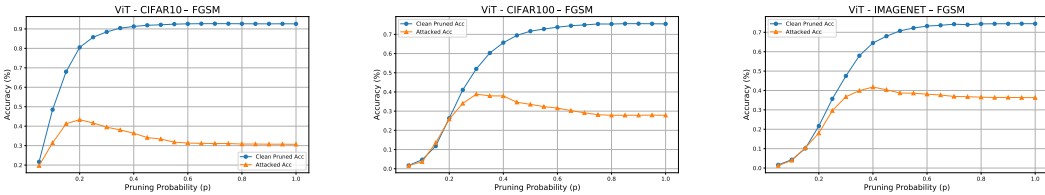

Figure 3: Clean and attacked accuracy of a pruned ViT when subject to the FGSM attack across varying pruning ratios for the CIFAR-10, CIFAR-100 and ImageNet datasets.

## 6.4 ON THE EFFECT OF MODEL'S SIZE

In the derived upper bounds, both for adversarial robustness and the optimality of the pruning, we observe a dependence on the model's size, denoted by the number of layers $L$. Specifically, in Theorem 2, the bound suggests an exponential relationship between $L$ and the expected robustness. To empirically validate this dependence, we evaluate three MLP architectures of increasing depth: a 2-layer "small" MLP, a 3-layer "medium" MLP, and a 4-layer "large" MLP. For each architecture, we apply both

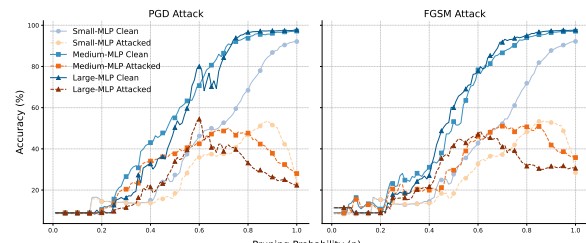

Figure 4: Effect of the model's size and the pruning parameters on both the clean and attacked accuracy.

FGSM and PGD attacks using identical training and attack configurations to ensure fair comparison. The results, shown in Figure 4, consistently reveal a trade-off between clean and adversarial accuracy as pruning increases. Notably, in larger models, achieving a favorable balance between clean and robust performance requires a higher pruning rate, thereby empirically supporting the influence of $L$.

## 6.5 PRACTICAL TRADE-OFF

In the previous section, we empirically demonstrated the existence of a trade-off between clean and adversarial accuracy under varying pruning probabilities. However, the central contribution of our work is the theoretical understanding of this relationship and specifically, how pruning parameters influence adversarial robustness (Theorem 2) and optimality (Proposition 1). These insights allow us

to formulate the problem as a bi-objective optimization task providing therefore a practical way of directly finding the best-trade off for a pre-trained model as detailed in Section 5, which we aim to validate experimentally. In this perspective, we consider two settings: (1) an MLP with a uniform pruning probability $p$ across layers, and (2) a layered setting with independent pruning probabilities $p^{(\ell)}$ per layer. For each, we vary the robustness–optimality trade-off parameter $\lambda$ and solve the optimization using the adaptation provided in Algorithm 1, as explained in details in Appendix E.

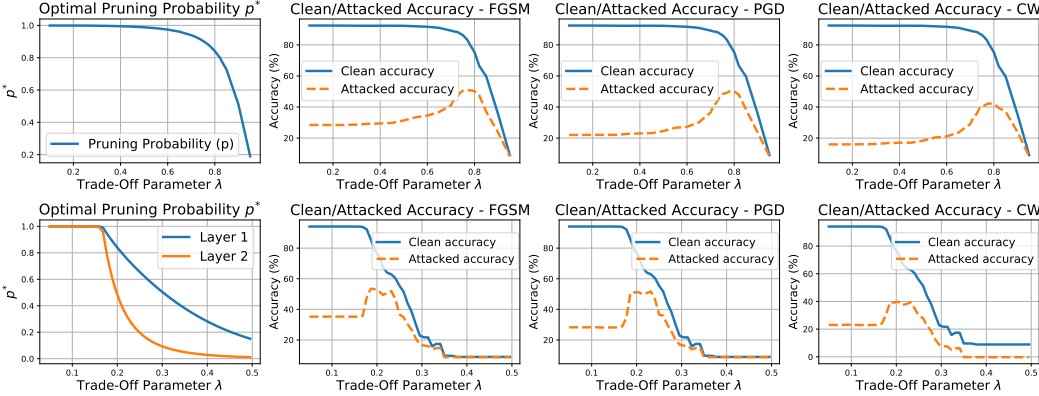

Figure 5: Optimal pruning and corresponding accuracies for uniform (top) and layer-wise (bottom) pruning. The optimization selects pruning levels in accordance with each layer contribution.

Figure 5 (top) shows the optimal pruning probability and corresponding clean and adversarial accuracy for the uniform pruning case, while Figure 5 (bottom) presents the results for layer-wise pruning. The resulting clean and adversarial accuracies align closely with our earlier empirical findings in which we considered different pruning values. Specifically, the findings confirm that solving the proposed optimization effectively selects pruning probabilities $p$ that yield the best trade-off based on the chosen penalization. Notably, the algorithm consistently prunes the second layer more than the first, highlighting a better clean-to-attacked accuracy ratio and indicating that different layers contribute unequally to robustness depending on their position in the network.

## 7 CONCLUSION

This work investigated the effect of pruning on the adversarial robustness of neural networks. We analyzed how specific pruning parameters influence robustness by deriving an upper bound that links these parameters to adversarial risk, thus providing guidance on selecting pruning configurations. However, tuning pruning parameters solely for robustness may harm the clean accuracy of the resulting model. To mitigate this, we also examined their impact on pruning optimality, ensuring the pruned model remains a close and faithful approximation of the original non-pruned model. By combining the two bounds, our study is the first to reveal a clear trade-off between clean and adversarial accuracy when subject to pruning, offering actionable insights for balancing both. With carefully chosen pruning parameters, we can improve robustness without additional constraints or cost, yielding a "free-lunch" benefit. Our empirical results confirm this trade-off across diverse models and datasets under various adversarial attack settings. This opens a promising avenue for future work on designing pruning strategies that are explicitly aware of adversarial robustness.

**Discussion.** In the current work, we have shown that pruning can yield additional benefits in terms of adversarial robustness while also affecting clean accuracy, thereby revealing a meaningful trade-off that can be controlled depending on the target application. Prior work by Guo et al. (2018) provided a theoretical justification for the robustness benefits of sparsity in the specific case of MLPs and also empirically demonstrated the existence of a robustness–accuracy trade-off. This empirical observation has been further supported by subsequent studies (Jordao & Pedrini, 2021; Piras et al., 2025), which have explored the trade-off through experimental analysis. Our work extends this line of research by analytically deriving how the trade-off is governed by the choice of pruning parameters. This leads to a principled optimization formulation that can guide the selection of pruning configurations to jointly enhance robustness and optimality, tailored to the use-case at hand.

ETHICS STATEMENT

This work does not involve human subjects and therefore does not require IRB approval. All datasets used are publicly available and appropriately licensed. Although adversarial attacks are employed, they are standard, publicly available methods used solely to evaluate and improve model robustness. In this context, our aim is to develop defense strategies that mitigate potential harm. To the best of our knowledge, this research does not raise ethical concerns related to discrimination, bias, privacy, or security. No conflicts of interest or legal compliance issues are associated with this work. We additionally note that LLMs were used only to assist with text refinement.

REPRODUCIBILITY STATEMENT

We have made an effort to ensure that our results can be reproduced by others. All datasets and pretrained models we use are publicly available and are clearly referenced in the paper. In addition, the code to reproduce our results is included in the Supplementary Materials and shall be made public upon publication. The experimental setup, including how the models are trained and how adversarial evaluations are carried out, is described in detail in the main text and the Appendix I. Additional proofs, derivations, and extended results are included in the appendix.

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

# Supplementary Material
# When Less Is More: Uncovering the Robustness Advantage of Model Pruning

## A    PROOF OF LEMMA 1

**Lemma.** *Let $f\colon \mathcal{X} \to \mathcal{Y}$ be the original TBM-based classifier following the considered problem setup. We have that the pruning strategy $\tau_{p=1}$ (i. e., no pruning) is $(\epsilon, \gamma)$-robust, with:*

$$\gamma = \left(\frac{d}{d-1}\right)^2 C_1 C_2 \epsilon,$$

*with*   $C_1 = \left(1 + \|W_O\|\sqrt{H} \max_h \left[\|W^{V,h}\| \left[\frac{4}{\sqrt{d/H}} \|W^{Q,h}\| \|W^{K,h}\| + 1\right]\right]\right)$ *and* $C_2 = \left(1 + \|W_{FFN}\|\right)$

*Proof.* Let's consider our input $X \in \mathcal{X}$ composed of $n$ tokens $x_i \in \mathbb{R}^d$. We consider that our model $f$ is built using the dot-product self-attention as referred to in Equation 1 and reformulated as:

$$\mathrm{AH}(x) = \mathrm{Softmax}\left(\frac{(XW^Q)(XW^K)^T}{\sqrt{\frac{D}{H}}}\right)XW^V$$
$$= PXW^V = h(X)W^V,$$

where $W^Q, W^K, W^V$ are learnable weights of the model. Let's consider the function $h(X)$, we can write:

$$f(X) = PX = \mathrm{Softmax}(XA^T X^T)X$$

$$f(X) = PX = \mathrm{Softmax}\left(XA^\top X^\top\right) X = \begin{bmatrix} h_1(X)^\top \\ \vdots \\ h_n(X)^\top \end{bmatrix} \in \mathbb{R}^{n \times d}, \quad \text{with:}$$

$$A = \frac{W^K W^{Q^\top}}{\sqrt{d/H}} \in \mathbb{R}^{d \times d} \quad \text{and} \quad h_i(X) = \sum_{j=1}^{n} P_{ij} x_j \quad \text{with} \quad P_i^\top = \mathrm{Softmax}(XAx_i).$$

By analyzing the partial derivatives, we can directly write the following regarding eh Jacobian matrix of $h$:

$$J_{ij} = X^\top P^{(i)} E_{ji} X A^\top + \delta_{ij} \left(X^\top P^{(i)} X A\right) + P_{ij} I_d,$$

with:

- $P^{(i)} = \mathrm{diag}\left(P_{i:}\right) - P_{i:}^\top P_{i:}$, [Softmax derivate]

- $E_{ji}$ is the $(n \times n)$ matrix with a single 1 in position $(j, i)$.

Based on this, two elements arises:

$$\text{If } i \neq j, \quad J_{ij} = X^\top P^{(i)} E_{ji} X A^\top + P_{ij} I, \tag{4}$$

$$\text{If } i = j, \quad J_{ii} = X^\top P^{(i)} E_{ii} X A^\top + X^\top P^{(i)}, X, A + P_{ii} I. \tag{5}$$

We recall that the input images are considered to be normalized, and therefore we can write:

$$\|X\| \leq 1$$

Additionally, since $P_{i:}$ is the output of the softmax, then can be considered a probability distribution. Therefore, $\sigma_{max}(diag(p)) \leq 1$ and $pp^T$ has rank 1:

$$\|P^{(i)}\| = \|\text{diag}(P_{i:}) - P_{i:}^\top P_{i:}\| \leq 2$$

**Case 1.** We start by considering the first case $i \neq j$, in which we have:

$$J_{ij} = X^\top P^{(i)} E_{ji} X A^\top + P_{ij} I.$$

Consequently we have the following:

$$\begin{aligned}
\|J_{ij}\| &\leq \|X^\top P^{(i)} E_{ji} X A^\top\| + \|P_{ij} I\| \\
&\leq 2 \times \|A\| + 1 \\
&\leq \|A\| + 1
\end{aligned}$$

**Case 2.** For the second case $i = j$, we have the following:

$$J_{ii} = X^\top P^{(i)} E_{ii} X A^\top + X^\top P^{(i)} X A + P_{ii} I.$$

We apply the same analogy as the previous case:

$$\begin{aligned}
\|J_{ii}\| &\leq \|X^\top P^{(i)} E_{ii} X A^\top\| + \|X^\top P^{(i)} X A\| + \|P_{ii} I\| \\
&\leq 2\|A\| + 2\|A\| + 1 \\
&\leq 4\|A\| + 1
\end{aligned}$$

So overall, we have the following:

$$\|J_{ij}\|_{op} \leq \begin{cases} 2\|A\| + 1, & \text{if } i \neq j, \\ 4\|A\| + 1, & \text{if } i = j. \end{cases}$$

So with our theoretical assumptions, the Jacobian is bounded and we have: $\mathcal{L}_h \leq 4\|A\| + 1$.

Specifically, for an attention head $h$, we have the following computation taking into account the different learnable weights:

$$\mathcal{L}_{head} \leq \|W^{V,h}\| \left[ \frac{4}{\sqrt{d/H}} \|W^{Q,h}\| \|W^{K,h}\| + 1 \right]$$

Since $f$ is represented by $H$ separate attention head, then their concatenated output as explained in Equation 2 is subject to the following:

$$\begin{aligned}
\mathcal{L}_{MH} &\leq \|W_O\| \sqrt{H} \max_h \left[ \mathcal{L}_{head} \right] \\
&\leq \|W_O\| \sqrt{H} \max_h \left[ \|W^{V,h}\| \left[ \frac{4}{\sqrt{d/H}} \|W^{Q,h}\| \|W^{K,h}\| + 1 \right] \right]
\end{aligned}$$

Finally, by applying the FFN and LN (with its parameters $\gamma = 1$ and $\beta = 1$), and since ReLU is 1-Lipschitz, we have the following result:

$$\begin{aligned}
\mathcal{L}_f &\leq L_{LN}^2 (1 + \mathcal{L}_{MH})(1 + L_{FFN}) \\
&\leq \left( \frac{d}{d-1} \right)^2 (1 + \mathcal{L}_{MH})(1 + \|W_{FFN}\|) \\
&\leq \left( \frac{d}{d-1} \right)^2 \left( 1 + \|W_O\| \sqrt{H} \max_h \left[ \|W^{V,h}\| \left[ \frac{4}{\sqrt{d/H}} \|W^{Q,h}\| \|W^{K,h}\| + 1 \right] \right] \right) (1 + \|W_{FFN}\|) \\
&\leq \left( \frac{d}{d-1} \right)^2 C_1 C_2,
\end{aligned}$$

$$\text{with} \quad C_1 = \big(1 + \|W_O\|\sqrt{H} \max_h \big[\|W^{V,h}\| \big[\frac{4}{\sqrt{d/H}}\|W^{Q,h}\|\|W^{K,h}\|+1\big]\big]\big)$$

$$C_2 = \big(1 + \|W_{FFN}\|\big)$$

Let's now consider a perturbed input $\tilde{x} \in \mathcal{B}(x, \epsilon)$ as defined in Section 4.1. The previous upper-bound applies to any given point within that budget, and therefore we have:

$$\sup_{\tilde{x} \in \mathcal{B}(x,\epsilon)} d_{\mathcal{Y}}(g(\tilde{x}), g(x)) \leq \mathcal{L}_f \epsilon$$

Since we consider that $p = 1$ (no pruning), then by taking into account the expectancy, we get the desired result.

$\square$

## B    PROOF OF THEOREM 1

**Theorem.** *Let $f \colon \mathcal{X} \to \mathcal{Y}$ to our original TBM-based classifier following our problem setup. Let $g$ be its corresponding pruned version using a pruning strategy $\tau_p$, then $\tau_p$ is $(\epsilon, \gamma) - robust$ with:*

$$\gamma' \leq C\gamma,$$

$$\text{with} \quad C = \frac{1 + p^2\|W^O\|_F \sqrt{H}\alpha}{1 + \|W^O\|_F \sqrt{H}\alpha} \times \frac{1 + p\|W_{FFN}\|_F}{1 + \|W_{FFN}\|_F} \leq 1$$

$$\text{and} \quad \alpha = \max_h \left[ \|W^{V,h}\|_F \left( \frac{4}{\sqrt{d/H}}\|W^{Q,h}\|_F\|W^{K,h}\|_F+1 \right) \right].$$

*Proof.* From the proof of Lemma 1 in Appendix A, we have the following results:

$$\mathcal{L}_f \leq \big(\frac{d}{d-1}\big)^2 C_1 C_2,$$

$$\text{with} \quad C_1 = \big(1 + \|W_O\|\sqrt{H} \max_h \big[\|W^{V,h}\| \big[\frac{4}{\sqrt{d/H}}\|W^{Q,h}\|\|W^{K,h}\|+1\big]\big]\big)$$

$$C_2 = \big(1 + \|W_{FFN}\|\big)$$

We consider that the model is pruned with a pruning strategy $\tau$ following the same analogy as the one provided in Section 3. We start by understanding the effect of such operation on the weight norm in terms of expectation. Let $W$ be our original weight, and let $B_{i,j}$ be the considered pruning mask (which is the realization of the pruning probability as explained in Section 3). We can write:

$$\mathbb{E}_\tau[\|W_g^{(\ell)}\|] = \mathbb{E}_\tau[\|B^{(\ell)} \odot W_f^{(\ell)}\|]$$

$$\leq \sqrt{\sum_{i,j} \mathbb{E}_\tau[B_{ij}^{(\ell)} W_{ij}^{(\ell)2}]}$$

$$\leq \sqrt{\sum_{i,j} p_{ij}^{(\ell)} W_{ij}^{(\ell)2}}$$

Since we consider pruning that uses the same parameter $p$, then we have:

$$\mathbb{E}[\|W_g^{(\ell)}\|] \leq \sqrt{p}\|W_f^{(\ell)}\|_F$$

Based on this, we can use the derived upper-bound and adapt accordingly:

$$C_1' \leq 1 + \|W_{(g)}^O\|_F \sqrt{H} \max_h \left[ \|W_{(g)}^{V,h}\|_F \left( \frac{4}{\sqrt{d/H}} \|W_{(g)}^{Q,h}\|_F \|W_{(g)}^{K,h}\|_F + 1 \right) \right] \tag{6}$$

$$\leq 1 + (p)^2 \|W^O\|_F \sqrt{H} \max_h \left[ \|W^{V,h}\|_F \left( \frac{4}{\sqrt{d/H}} \|W^{Q,h}\|_F \|W^{K,h}\|_F + 1 \right) \right] \tag{7}$$

$$= 1 + p^2 \|W^O\|_F \sqrt{H} \alpha, \tag{8}$$

where we define:

$$\alpha = \max_h \left[ \|W^{V,h}\|_F \left( \frac{4}{\sqrt{d/H}} \|W^{Q,h}\|_F \|W^{K,h}\|_F + 1 \right) \right]. \tag{9}$$

And similarly, for the second constant:

$$C_2' = 1 + \|W_{\text{FFN},(g)}\| \leq 1 + p\|W_{\text{FFN}}\|_F. \tag{10}$$

Similarly, when considering the original model $f$, since the spectral norm is always smaller than the frobenius norm, we can re-write the $C_1$ and $C_2$ accordingly.

Let's now consider the difference between the two terms for both $f$ and $g$:

$$C = \frac{C_1' C_2'}{C_1 C_2} = \frac{(1 + p^2 \|W^O\|_F \sqrt{H} \alpha)(1 + p\|W_{\text{FFN}}\|_F)}{(1 + \|W^O\|_F \sqrt{H} \alpha)(1 + \|W_{\text{FFN}}\|_F)}. \tag{11}$$

Thus, the final bound on the Lipschitz constant of the pruned model is:

$$\gamma' \leq C\gamma, \tag{12}$$

where:

$$C = \frac{1 + p^2 \|W^O\|_F \sqrt{H} \alpha}{1 + \|W^O\|_F \sqrt{H} \alpha} \times \frac{1 + p\|W_{\text{FFN}}\|_F}{1 + \|W_{\text{FFN}}\|_F} \leq 1. \tag{13}$$

and:

$$\alpha = \max_h \left[ \|W^{V,h}\|_F \left( \frac{4}{\sqrt{d/H}} \|W^{Q,h}\|_F \|W^{K,h}\|_F + 1 \right) \right]. \tag{14}$$

$\square$

## C  PROOF OF THEOREM 2

**Theorem.** *Let $f : \mathcal{X} \to \mathcal{Y}$ be an L-Layers MLP classifier. Let $g$ be its corresponding pruned version obtained through our considered pruning strategy with probabilities $p_{i,j}^{(\ell)}$. Then the pruning strategy $\tau$ is $(\epsilon, \gamma)$-robust with:*

$$\gamma = P_L \prod_{\ell=1}^{L} \|W_f^{(\ell)}\|_F, \text{ with } \quad P_L = \prod_{\ell=1}^{L} \sqrt{\max_{i,j} p_{ij}^{(\ell)}}$$

*Proof.* We start from a classifier $f : \mathcal{X} \to \mathcal{Y}$, with its corresponding weights denoted as $W^\ell$ and its corresponding pruned version $g$, with its weights denoted as $W'^{(\ell)}$. We consider a probabilistic pruning approach $\tau$ as discussed in Section 3, where each weight $W_{i,j}^{(\ell)}$ is independently pruned using a Bernoulli distribution with probability $p_{i,j}^{(\ell)}$. Hence,

$$W'^{(\ell)}_{i,j} = \begin{cases} W_{i,j}^{(\ell)} & \text{with probability } p_{i,j}^{(\ell)}, \\ 0, & \text{with probability } 1 - p_{i,j}^{(\ell)}. \end{cases}$$

For each layer $\ell \leq L$, the pruning can be formulate as the following:

$$W'^{(\ell)} = B^{(\ell)} \odot W^{(\ell)}, \qquad B_{ij}^{(\ell)} \sim \text{Ber}\big(p_{ij}^{(\ell)}\big). \qquad (1)$$

Similar to the previous proof, for each individual weight $\ell$, considering the linearity of the expected value, we have in expectancy:

$$\mathbb{E}_\tau\big[\|B^{(\ell)} \odot W^{(\ell)}\|\big] \leq \mathbb{E}_\tau\big[\|B^{(\ell)} \odot W^{(\ell)}\|_F\big]$$

$$\leq \mathbb{E}_\tau\Big[\sqrt{\sum_{i,j} B_{ij}^{(\ell)} W_{ij}^{(\ell)2}}\Big]$$

$$\leq \sqrt{\sum_{i,j} \mathbb{E}\big[B_{ij}^{(\ell)} W_{ij}^{(\ell)2}\big]}$$

$$\leq \sqrt{\sum_{i,j} p_{ij}^{(\ell)} W_{ij}^{(\ell)2}}$$

$$\leq \sqrt{\max_{i,j} p_{ij}^{(\ell)}} \sqrt{\sum_{i,j} W_{ij}^{(\ell)2}}$$

$$\leq \sqrt{\max_{i,j} p_{ij}^{(\ell)}} \|W^{(\ell)}\|_F$$

For the model $g$, we know that:

$$\mathbb{E}_\tau\big[\|g(x) - g(x')\|\big] \leq \mathbb{E}_\tau\Big[\prod_{\ell=1}^{L} \|W_g^{(\ell)}\|\Big]$$

$$\leq \prod_{\ell=1}^{L} \|\sqrt{\max_{i,j} p_{ij}^{(\ell)}} \|W_f^{(\ell)}\|_F$$

$$\leq P_L \prod_{\ell=1}^{L} \|W_f^{(\ell)}\|_F$$

with:

$$P_L = \prod_{\ell=1}^{L} \sqrt{\max_{i,j} p_{ij}^{(\ell)}}$$

$\square$

## D    PROOF OF PROPOSITION 1

**Proposition.** *Let $f$ be a $L$-layer MLP and $g$ be its corresponding pruned version obtained through our considered pruning strategy with probabilities $p_{i,j}^{(\ell)}$. For an input point $x_0 \in \mathcal{X}$, the chosen pruning strategy is $\zeta$-optimal, with:*

$$\zeta = \|x_0\| \prod_{\ell=1}^{L} \|W^{(\ell)}\| \sum_{\ell=1}^{L} \frac{1}{\|W^{(\ell)}\|} \sqrt{\sum_{i,j} (1 - p_{i,j}^{(\ell)})\big(W_{i,j}^{(\ell)}\big)^2}.$$

*Proof.* Let $f$ be a MLP of $L$ layers with 1-Lipschitz activation functions (such as ReLu and TanH). We additionally consider a Bernoulli-like pooling such as the one provided in Section 4.1 and which can be written as:

$$\hat{W}_{i,j}^{(l)} = \begin{cases} W_{i,j}^{(\ell)} & \text{with probability } p_{i,j}^{(\ell)}, \\ 0, & \text{with probability } 1 - p_{i,j}^{(\ell)}. \end{cases}$$

We define the following quantity:

$$\Delta^{(l)} = \hat{W}^{(l)} - W^{(l)}, \text{ with } l = 1, \ldots, L.$$

We can consequently write:

$$\Delta_{i,j}^{(l)} = \begin{cases} 0, & \text{with probability } p_{i,j}^{(\ell)}, \\ -W_{i,j}^{(l)} & \text{otherwise.} \end{cases}$$

**Part 1:** Let's consider the model $f$, since it's a MLP, we can write the following:

$$f(x) = x^{(L)} = W^{(L)}\sigma^{(L)}(x^{(L-1)})$$
$$= W^{(L)}\sigma^{(L)}(W^{(L-1)}\sigma^{(L-1)}(x^{(L-2)}))$$

and similarly:

$$g(x) = x'^{(L)} = \hat{W}^{(L)}\sigma^{(L)}(x'^{(L-1)})$$
$$= (W^{(L)} + \Delta^{(L)})\sigma^{(L)}(x'^{(L-1)})$$

We therefore write:

$$\|f(x) - g(x)\| = \|W^{(L)}(\sigma^{(L)}(x^{(L-1)}) - \sigma^{(L)}(x'^{(L-1)})) + \Delta^{(L)}\sigma^{(L)}(x'^{(L-1)})\|$$
$$\leq \|W^{(L)}\|\|x^{(L-1)} - x'^{(L-1)}\| + \|\Delta^{(L)}\|\sigma^{(L)}(x'^{(L-1)})\|$$

We also have by recursion the following (since $\sigma$ is 1-Lipschitz, and by taking $x$ and 0):

$$\|\Delta^{(l)}\|\|\sigma^{(l)}(x'^{(l-1)})\| \leq \|x_0\|\|\Delta^{(L)}\|\prod_{j=1}^{L-1}\|W^{(j)}\|.$$

Note that we directly use $W^{(j)}$ in the previous inequality rather than $\hat{W}^{(j)}$ since by definition the original weight is always upper-bounding in terms of norm the pruned weight.

Combining the two inequalities and by recursive iteration again, we find:

$$\|f(x) - g(x)\| \leq \|W^{(L)}\|\|x^{(L-1)} - x'^{(L-1)}\| + \|x_0\|\|\Delta^{(L)}\|\prod_{j=1}^{l-1}\|W^{(j)}\| \tag{15}$$

$$\leq \|x_0\|\prod_{i=1}^{L}\|W^{(i)}\|\sum_{i=1}^{L}\frac{\|\Delta^{(i)}\|}{\|W^{(i)}\|}. \tag{16}$$

We note that $\Delta^{(\ell)}$ is a random matrix, following the Bernoulli distribution, hence by taking the expectation on both sides, we get:

$$\mathbb{E}_{\mathcal{P}}\left[\|f(x) - g(x)\|\right] \leq \|x_0\|\prod_{\ell=1}^{L}\|W^{(\ell)}\|\sum_{\ell=1}^{L}\frac{1}{\|W^{(\ell)}\|}\mathbb{E}\left[\|\Delta^{(\ell)}\|\right].$$

We additionally have the following:

$$\mathbb{E}\left[\|\Delta^{(\ell)}\|\right] \leq \mathbb{E}\left[\|\Delta^{(\ell)}\|_F\right] \leq \sqrt{\mathbb{E}\left[\|\Delta^{(\ell)}\|_F^2\right]},$$

where $\|\cdot\|_F$ is the Frobenius norm, with:

$$\|\Delta^{(\ell)}\|_F^2 = \sum_{i,j}(\Delta_{i,j}^{(\ell)})^2$$
$$= \sum_{i,j}(1 - p_{i,j}^{(\ell)})(W_{i,j}^{(\ell)})^2.$$

Combining everything, we get:

$$\mathbb{E}\big[\|\Delta^{(\ell)}\|\big] \leq \sqrt{\sum_{i,j}(1 - p_{i,j}^{(\ell)})(W_{i,j}^{(\ell)})^2}.$$

And consequently, we get:

$$\mathbb{E}_{\mathcal{P}}\Big[\big\|f(x) - g(x)\big\|\Big] \leq \|x_0\| \prod_{\ell=1}^{L}\|W^{(\ell)}\| \sum_{\ell=1}^{L}\frac{1}{\|W^{(\ell)}\|}\mathbb{E}\big[\|\Delta^{(\ell)}\|\big]$$

$$\leq \|x_0\| \prod_{\ell=1}^{L}\|W^{(\ell)}\| \sum_{\ell=1}^{L}\frac{1}{\|W^{(\ell)}\|}\sqrt{\sum_{i,j}(1 - p_{i,j}^{(\ell)})(W_{i,j}^{(\ell)})^2}.$$

$\square$

# E  ON THE PRACTICAL ASPECT OF THE TRADE-OFF

In Section 5.3, we pointed out to the existence of the trade-off and its formalization as a bi-objective optimization problem as follows:

$$\min_p \quad \lambda\gamma(p_{i,j}^{(\ell)}) + (1 - \lambda)\zeta(p_{i,j}^{(\ell)}) \quad \text{s.t.} \quad 0 \leq p_{i,j}^{(\ell)} \leq 1.$$

The parameter $\lambda$ governs the trade-off between the optimality and robustness quantity.

In this section, we aim to analyze this objective and come-up with a practical optimization plan to solve it, providing therefore an insightful application of the trade-off in finding the best parameters to find relevant pruning probabilities that satisfy a user's desire to optimize optimality and adversarial robustness.

In this specific setting, we focus on the case of uniform pruning probability for each layer. From Theorem 2, by considering $p^{(\ell)}$ as the corresponding pruning probability at layer $\ell$, we have:

$$\gamma(p) = \left(\prod_{\ell=1}^{L}\|W^{(\ell)}\|_F\right)\prod_{\ell=1}^{L}\sqrt{p_\ell} = A\prod_{\ell=1}^{L}\sqrt{p_\ell}, \tag{17}$$

where $A = \prod_{\ell=1}^{L}\|W^{(\ell)}\|_F$ is the product of the weight norms of the considered fixed pretrained network. In addition, using Proposition 1, with the same uniform $p_\ell$ we get the following bound on the optimality:

$$\zeta(p) = \mathbb{E}\|x\| \prod_{\ell=1}^{L}\|W^{(\ell)}\|_2 \sum_{\ell=1}^{L}\frac{\|W^{(\ell)}\|_F}{\|W^{(\ell)}\|_2}\sqrt{1 - p_\ell} \tag{18}$$

$$= C\sum_{\ell=1}^{L}b_\ell\sqrt{1 - p_\ell} \tag{19}$$

where $C = \mathbb{E}\|x\| \prod_{\ell=1}^{L}\|W^{(\ell)}\|_2$ and $b_\ell = \frac{\|W^{(\ell)}\|_F}{\|W^{(\ell)}\|_2}$. We aim to minimize the special case of the scalarized objective:

$$J(p) = \lambda\gamma(p) + (1 - \lambda)\zeta(p), \text{ s.t. } 0 \leq p_\ell \leq 1, \quad \lambda \in [0, 1]. \tag{20}$$

We additionally can write the following:

$$\frac{\partial}{\partial p_k}\gamma(p) = \frac{\gamma(p)}{2p_k} \text{ and } \frac{\partial}{\partial p_k}\zeta(p) = -\frac{C\,b_k}{2\sqrt{1 - p_k}}$$

Therefore differentiating Equation 20 by setting $\frac{\partial J}{\partial p_k} = 0$ gives the following:

$$\frac{\lambda\gamma(p)}{p_k} = \frac{(1-\lambda)Cb_k}{\sqrt{1-p_k}}. \tag{21}$$

Let's denote $t = \sqrt{1-p_k} \in [0,1]$ so then we have: $p_k = 1 - t^2$. Let's now substitute using the previous quantity in Equation 21, we get the following nice quandratic format:

$$(1-\lambda)Cb_k t^2 + \lambda\gamma(p)t - (1-\lambda)Cb_k = 0, \tag{22}$$

And typically its nonnegative root can be formulated as:

$$t^\star = \frac{-\lambda\gamma(p) + \sqrt{\left(\lambda\gamma(p)\right)^2 + 4(1-\lambda)^2C^2b_k^2}}{2(1-\lambda)Cb_k}. \tag{23}$$

The coordinate descent individual update at an iteration $k$ can be therefore formulated as:

$$p_k = 1 - \left(t^\star\right)^2, \tag{24}$$

We recall that since $\gamma(p)$ in Equation 23 uses the product of weight norms, so Equation 24 is actually defining a fixed point/coordinate-descent iteration.

In practice, given the previous derivation of the update, we can directly use the coordinate-descent algorithm to solve the problem. The algorithm is what follows.

---

**Algorithm 1** Coordinate-Descent Application for optimal Pruning Probabilities

---

**Require:** Model Weights $\{W^{(\ell)}\}_{\ell=1}^L$, trade-off $\lambda \in [0,1]$, Optimization tolerance $\varepsilon > 0$, Optimization Max Iterations $T_{\max}$

1: Compute Constant Values : $A = \prod_{\ell=1}^L \|W^{(\ell)}\|_F$; $\quad b_\ell = \|W^{(\ell)}\|_F / \|W^{(\ell)}\|_2$ for $\ell = 1 \dots L$ ; $C = \left(\mathbb{E}\|x\|\right)\prod_{\ell=1}^L \|W^{(\ell)}\|_2$

2: Initialize $p_\ell^{(0)} \in [0,1]$

3: $J^{(0)} = \lambda A \prod_\ell \sqrt{p_\ell^{(0)}} + (1-\lambda)C \sum_\ell b_\ell \sqrt{1 - p_\ell^{(0)}}$

4: **for** $t = 0, 1, \dots, T_{\max} - 1$ **do**

5: $\quad \gamma = A \prod_{\ell=1}^L \sqrt{p_\ell^{(t)}}$

6: $\quad$ **for** $k = 1$ **to** $L$ **do**

7: $\quad\quad t_k^\star \leftarrow \dfrac{-\lambda\gamma + \sqrt{(\lambda\gamma)^2 + 4(1-\lambda)^2C^2b_k^2}}{2(1-\lambda)Cb_k}$

8: $\quad\quad p_k^{(t+1)} \leftarrow \min\{1, \max\{0, 1 - (t_k^\star)^2\}\}$

9: $\quad$ **end for**

10: $\quad J^{(t+1)} = \lambda A \prod_\ell \sqrt{p_\ell^{(t+1)}} + (1-\lambda)C \sum_\ell b_\ell \sqrt{1 - p_\ell^{(t+1)}}$

11: $\quad$ **if** $\dfrac{\left|J^{(t+1)} - J^{(t)}\right|}{J^{(t)} + 10^{-12}} < \varepsilon$ **then**

12: $\quad\quad$ **break**

13: $\quad$ **end if**

14: **end for**

15: **return** $p^\star \leftarrow \{p_\ell^{(t+1)}\}_{\ell=1}^L$

---

In practice, we have seen that setting the number of iterations between 20 and 50 was more than enough to reach satisfactory convergence results. The optimization results are provided in the Experimental evaluation (Section 6). Empirically, we have seen that the algorithm converges in a very limited time and therefore doesn't need a large complexity and overhead.

**Complexity of the algorithm.** The complexity of the coordinate descent algorithm for optimizing $J$ is directly proportional to the number of layers $L$, as each iteration involves updating $p_k$ for all

$k = 1, 2, \ldots, L$. For each layer, the update step requires computing a product over $L - 1$ terms and solving a simple closed-form expression, making the per-layer computational cost very low. Consequently, the overall complexity of one full iteration is $\mathcal{O}(L)$ (Wright, 2015), which scales linearly with the number of layers. This computational cost is negligible compared to the overall complexity. Therefore, the coordinate descent algorithm adds minimal overhead.

## F    ON THE EXISTENCE OF THE PARETO-OPTIMALITY

The bi-objective optimization problem formulated in Section 5.2 naturally leads to the question: *what is the complete set of optimal trade-offs between robustness and optimality?*. In this perspective, we aim to address this question through a Pareto-optimality framework.

**Definition 3** (Pareto-Optimality). *A pruning configuration $p^* = \{p_{i,j}^{(\ell)*}\}_{\ell,i,j}$ is Pareto-optimal if there exists no alternative $p'$ satisfying $\gamma(p') \leq \gamma(p^*)$ and $\zeta(p') \leq \zeta(p^*)$ with at least one strict inequality. The set of all Pareto-optimal configurations forms the* Pareto front *in $(\gamma, \zeta)$ space.*

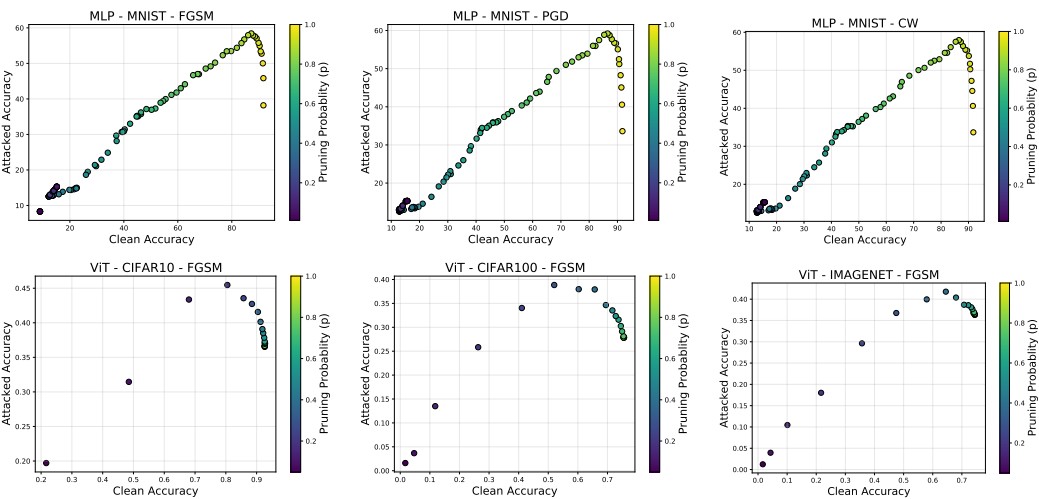

Figure 6: Empirical Pareto fronts in clean vs. attacked accuracy space. Each point represents a different pruning probability $p \in [0, 1]$, color-coded by $p$ value. Points on the approximate upper-right boundary represent empirically Pareto-optimal trade-offs where neither accuracy metric can be improved without degrading the other. (Top) MLP-MNIST under different attacks (FGSM, PGD, CW), showing attack-specific but qualitatively similar frontiers. (Bottom) ViT on CIFAR-10, CIFAR-100, and ImageNet-100, demonstrating how dataset complexity affects achievable trade-offs. The non-monotonic relationship between pruning probability and attacked accuracy reflects the interplay between improved stability (captured by $\gamma$) and degraded model capacity (captured by $\zeta$).

The provided accuracy plots in which we consider the variation of the clean and attacked accuracy based on varying the value of pruning probability $p$ (Figures 2, 3 and Figures 8-10-13-14) implicitly showed the existence of Pareto fronts. Furthermore, Figure 6 reformulates these results to directly visualize empirical Pareto frontiers in accuracy space: points on the approximate upper-right boundary represent configurations where neither clean nor attacked accuracy can be simultaneously improved. Notably, the curves exhibit non-monotonic behavior, as attacked accuracy typically peaks at moderate pruning levels before degrading under aggressive pruning, highlighting the "sweet spots" identified in Section 6.3 as specific empirically Pareto-optimal points.

Beyond empirical validation, we aim in this section to analyze theoretically the existence of such Pareto effect, specifically for the special case of uniform pruning for MLP.

**Lemma 2** (Pareto Front for Uniform Pruning). *Under uniform pruning with $p_{i,j}^{(\ell)} = p \in [0,1]$, the Pareto front in $(\gamma, \zeta)$ space is parameterized by:*

$$\gamma(p) = p^{L/2} \prod_{\ell=1}^{L} \|W_f^{(\ell)}\|_F,$$

$$\zeta(p) = \|x_0\| \prod_{\ell=1}^{L} \|W^{(\ell)}\| \cdot \sqrt{1-p} \sum_{\ell=1}^{L} \frac{\|W^{(\ell)}\|_F}{\|W^{(\ell)}\|},$$

*for $p \in [0,1]$, and the **trade-off rate between objectives satisfies** $\mathbf{d\gamma/d\zeta < 0}$.*

The trade-off rate $d\gamma/d\zeta < 0$ confirms that in the theoretical bound space, improving one bound necessarily degrades the other, with depth $L$ amplifying this effect. We note that while Theorem 2 and Proposition 1 hold for fully general layer-wise pruning with independent $p_{i,j}^{(\ell)}$, the closed-form parametric expressions above require uniform pruning for analytical tractability. Full derivation and necessary conditions for layer-wise pruning are in Appendix G.

**Relationship between theoretical bounds and empirical accuracy.** While Lemma 2 characterizes the Pareto front in $(\gamma, \zeta)$ bound space, Figure 6 shows the empirical front in accuracy space. These are related but distinct: attacked accuracy depends on both output stability (captured by $\gamma$) and base model quality (captured by $\zeta$). At moderate pruning ($p \approx 0.85$–$0.9$), both bounds remain controlled, yielding high attacked accuracy at the empirical "sweet spots". Under aggressive pruning, $\gamma$ continues to improve but $\zeta$ degrades substantially, explaining why attacked accuracy eventually declines. Our scalarized optimization balances both bounds, naturally identifying configurations that perform well on task-specific metrics.

**Connection to optimization and practical implications.** Figure 5, where we varied $\lambda$ to solve the scalarized problem, implicitly performed a Pareto front sweep in bound space: each $\lambda$ selects a different trade-off point according to preference weights. The scalarization parameter therefore encodes the decision-maker's robustness-versus-optimality priority. Figure 6 enables empirical comparison across architectures and attacks, suggesting that score-based pruning (ViT, bottom row) may achieve more favorable trade-offs than magnitude-based methods (MLP, top row), though direct comparison requires consistent experimental conditions. More importantly, the Pareto framework clarifies that no single "optimal" pruning exists: the choice depends on application-specific priorities. Our $\lambda$-parameterized optimization provides a principled navigation method, allowing practitioners to select configurations aligned with their requirements, while Lemma 2 provides formal guarantees on bounds.

**Non-convexity and local optimality.** The non-convex objectives $\gamma(p)$ and $\zeta(p)$ over the high-dimensional space $p = \{p_{i,j}^{(\ell)}\}$ imply that our coordinate descent optimization converges to locally Pareto-optimal points. Globally optimal configurations may exist but require large parameter jumps. Our scalarization across varying $\lambda$ depicted in Figure 5 serves as a multi-start strategy; the diversity of solutions suggests successful exploration, though global optimality cannot be guaranteed. Additionally, the Pareto front may exhibit disconnected segments or sharp corners from structural changes (e.g., entire layers eliminated). Therefore, if the front has concave regions, a linear scalarization cannot generate points there. Our empirical fronts appear largely convex/mildly concave, suggesting scalarization is adequate for these architectures. Finally, practical pruning rounds continuous probabilities to binary masks, yielding discrete point sets rather than smooth frontiers, as reflected in our plots. Characterizing discretization effects and designing provably optimal discrete strategies remain open problems.

In short, our computed Pareto fronts represent locally optimal trade-offs in bound space that translate to empirically effective configurations in accuracy space. While multi-start scalarization mitigates non-convexity, alternative globally optimal configurations may exist beyond current optimization reach.

# G  PARETO-OPTIMALITY: ADDITIONAL RESULTS AND PROOFS

In this section, we provide the complete derivations for the Pareto-optimality framework.

### G.1 PROOF OF LEMMA 2

**Lemma 3** (Pareto Front for Uniform Pruning). *Under uniform pruning with $p_{i,j}^{(\ell)} = p \in [0,1]$, the Pareto front in $(\gamma, \zeta)$ space is parameterized by:*

$$\gamma(p) = p^{L/2} \prod_{\ell=1}^{L} \|W_f^{(\ell)}\|_F,$$

$$\zeta(p) = \|x_0\| \prod_{\ell=1}^{L} \|W^{(\ell)}\| \cdot \sqrt{1-p} \sum_{\ell=1}^{L} \frac{\|W^{(\ell)}\|_F}{\|W^{(\ell)}\|},$$

*for $p \in [0,1]$. The trade-off rate between objectives satisfies:*

$$\frac{d\gamma}{d\zeta} = -Lp^{(L/2)-1}\sqrt{1-p} \cdot \frac{\prod_{\ell=1}^{L} \|W_f^{(\ell)}\|_F}{\|x_0\| \prod_{\ell=1}^{L} \|W^{(\ell)}\| \sum_{\ell=1}^{L} \frac{\|W^{(\ell)}\|_F}{\|W^{(\ell)}\|}}.$$

*Proof.* From Theorem 2, we have:

$$\gamma(p) = P_L \prod_{\ell=1}^{L} \|W_f^{(\ell)}\|_F, \quad \text{where} \quad P_L = \prod_{\ell=1}^{L} \sqrt{\max_{i,j} p_{ij}^{(\ell)}}.$$

Under uniform pruning, $p_{i,j}^{(\ell)} = p$ for all $\ell, i, j$, hence $\max_{i,j} p_{ij}^{(\ell)} = p$ and:

$$P_L = \prod_{\ell=1}^{L} \sqrt{p} = p^{L/2}.$$

This immediately gives $\gamma(p) = p^{L/2} \prod_{\ell=1}^{L} \|W_f^{(\ell)}\|_F$.

From Proposition 1, we have:

$$\zeta = \|x_0\| \prod_{\ell=1}^{L} \|W^{(\ell)}\| \sum_{\ell=1}^{L} \frac{1}{\|W^{(\ell)}\|} \sqrt{\sum_{i,j}(1 - p_{i,j}^{(\ell)})(W_{i,j}^{(\ell)})^2}.$$

Under uniform pruning, $(1 - p_{i,j}^{(\ell)}) = (1 - p)$ for all $i, j, \ell$, thus:

$$\sqrt{\sum_{i,j}(1 - p_{i,j}^{(\ell)})(W_{i,j}^{(\ell)})^2} = \sqrt{1-p}\sqrt{\sum_{i,j}(W_{i,j}^{(\ell)})^2} = \sqrt{1-p}\|W^{(\ell)}\|_F.$$

Substituting:

$$\zeta(p) = \|x_0\| \prod_{\ell=1}^{L} \|W^{(\ell)}\| \cdot \sqrt{1-p} \sum_{\ell=1}^{L} \frac{\|W^{(\ell)}\|_F}{\|W^{(\ell)}\|}.$$

To compute the trade-off rate, we differentiate both parametric expressions with respect to $p$:

$$\frac{d\gamma}{dp} = \frac{L}{2}p^{(L/2)-1} \prod_{\ell=1}^{L} \|W_f^{(\ell)}\|_F,$$

$$\frac{d\zeta}{dp} = \|x_0\| \prod_{\ell=1}^{L} \|W^{(\ell)}\| \cdot \frac{-1}{2\sqrt{1-p}} \sum_{\ell=1}^{L} \frac{\|W^{(\ell)}\|_F}{\|W^{(\ell)}\|}.$$

By the chain rule:

$$\frac{d\gamma}{d\zeta} = \frac{d\gamma/dp}{d\zeta/dp} = -Lp^{(L/2)-1}\sqrt{1-p} \cdot \frac{\prod_{\ell=1}^{L} \|W_f^{(\ell)}\|_F}{\|x_0\| \prod_{\ell=1}^{L} \|W^{(\ell)}\| \sum_{\ell=1}^{L} \frac{\|W^{(\ell)}\|_F}{\|W^{(\ell)}\|}}.$$

Note that $d\gamma/d\zeta < 0$ for all $p \in (0,1)$, confirming the trade-off relationship. $\square$

## G.2 Necessary Conditions for Layer-Wise Pareto-Optimality

For the general case of layer-wise pruning with independent $p_{i,j}^{(\ell)}$, we provide necessary conditions:

**Remark 1** (Necessary Conditions for Pareto-Optimality). *If $p^* = \{p_{i,j}^{(\ell)*}\}_{\ell,i,j}$ is Pareto-optimal, then there exist non-negative constants $\mu_\gamma, \mu_\zeta \geq 0$ (not both zero) such that:*

$$\mu_\gamma \frac{\partial \gamma}{\partial p_{i,j}^{(\ell)}}\bigg|_{p^*} + \mu_\zeta \frac{\partial \zeta}{\partial p_{i,j}^{(\ell)}}\bigg|_{p^*} = 0,$$

*for all layers $\ell$ and weights $(i,j)$ where $p_{i,j}^{(\ell)*} \in (0,1)$, with appropriate complementarity conditions at boundaries.*

This follows from standard KKT conditions for multi-objective optimization (Boyd & Vandenberghe, 2004). At any Pareto-optimal point, the gradient of a weighted sum of objectives must vanish for some non-negative weights.

The constants $\mu_\gamma$ and $\mu_\zeta$ represent the marginal trade-off weights at that frontier point. Notably, our scalarization parameter $\lambda$ corresponds to $\mu_\gamma = \lambda$, $\mu_\zeta = 1 - \lambda$, confirming that varying $\lambda$ traces the Pareto front.

# H Additional Empirical Results

## H.1 Estimation Of Adversarial Risk And Optimality

In addition to the provided results in the main paper, we applied the same analysis to other datasets for each model. Specifically, for the CNN we considered the CIFAR-100 and for the ViT we provided the CIFAR-10 covering therefore all the datasets for these models. Figure 7 provides the results of the study. We see that similar insights as the one provided in the main paper are seen, validating therefore the existence of our discussed trade-off.

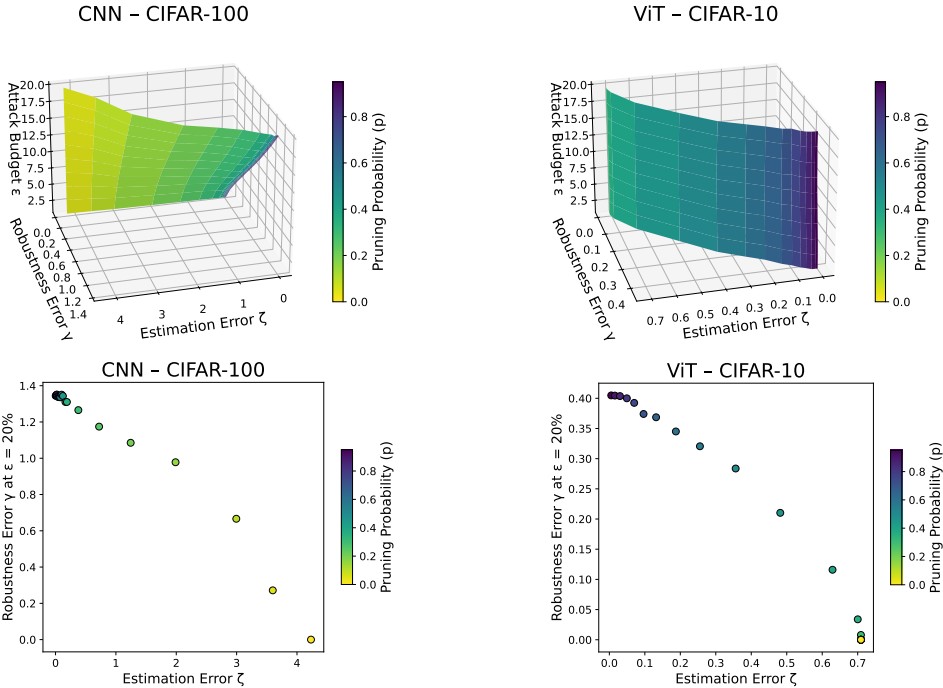

Figure 7: Additional results on the empirical analysis of the effect of the pruning parameters on the adversarial risk (Definition 1) and the Estimation Error (Equation 2) when subject to a range of attack budget ($\epsilon$).

## H.2   COMPLETE RESULTS - MLP

As we previously mentioned, in our analysis we focused on both the FGSM (provided in the main paper) and the PGD adversarial attack. In this context, Figure 8 provides the analysis on our considered MLP model for both the FGSM and PGD using both MNIST and CIFAR-10 Dataset. We can see that similar insights are seen for both these datasets. Specifically, the existence of the trade-off and a sweet spot in which the balance between adversarial and clean accuracy is interesting.

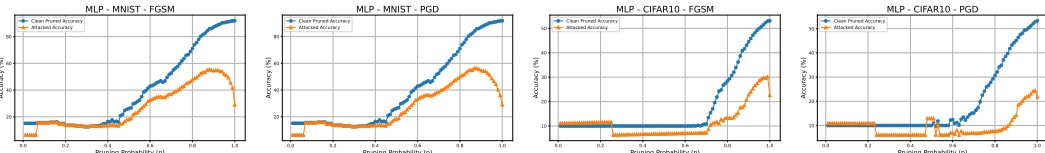

Figure 8: Clean and attacked accuracy of a pruned MLP model when subject to FGSM and PGD adversarial attacks and different pruning probabilities.

## H.3   COMPLETE RESULTS - CNN

In line with the previous section, we also extended the study of the CNN to the PGD attack. Figure 9 provides such analysis where we can see again similar insights as the one provided in the case of MLP. Specifically, as we decrease the pruning probability (making the model sparser), the attacked accuracy start going up, showcasing an enhancement in the adversarial robustness of the model, before decreasing due to both the optimality of the pruning strategy and the attack itself.

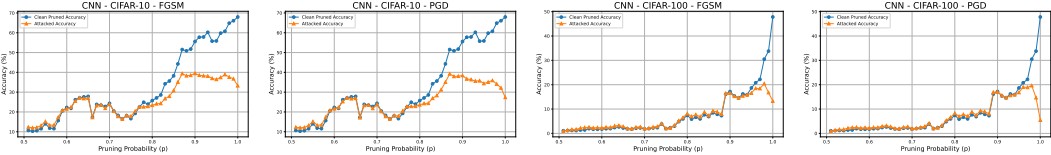

Figure 9: Clean and attacked accuracy of a pruned CNN model when subject to FGSM and PGD adversarial attacks and different pruning probabilities.

## H.4   COMPLETE RESULTS - VIT

Similar to the other models, in the main paper we only consider the FGSM attack when considering the ViT model. We therefore report the results using the PGD attack. We recall that for the ViT, we are rather considering a score-based pruning strategy. Figure 10 provides the resulting results of the study.

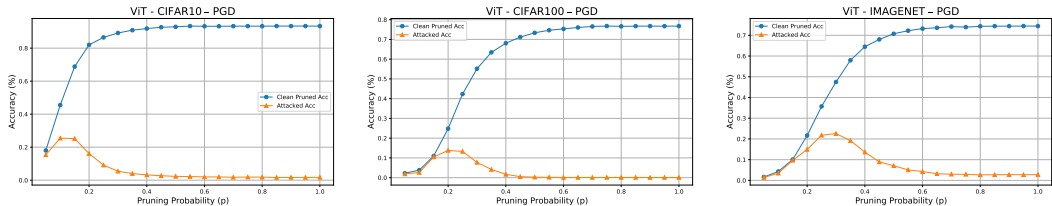

Figure 10: Clean and adversarial accuracy of pruned models under PGD attack across varying pruning probabilities.

We note that in previous experiments we have chosen to set the attack budget to $\epsilon = 4/255$, nonetheless, our theoretical analysis shall be applicable to any $\epsilon$. To further validate this, we computed the same analysis for $\epsilon = 8/255$ in the case of the FGSM under the ViT model.

Figure 11 provides the results of the analysis, where we see the same pattern of the effect pruning and in which we observe the existence of the trade-off robustness/optimality.

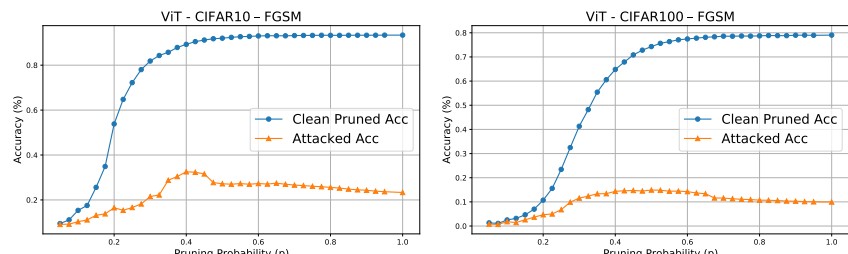

Figure 11: Clean and adversarial accuracy of pruned models under FGSM with an attack budget $\epsilon = 8/255$ across varying pruning probabilities.

## H.5 ADDITIONAL RESULTS - CW ATTACK AND AUTOATTACK

Beyond FGSM and PGD, we evaluate CW Attack (Carlini & Wagner, 2017), which uses targeted optimization for lower-distortion perturbations, and AutoAttack (Croce & Hein, 2020), a parameter-free ensemble providing reliable worst-case robustness estimates.

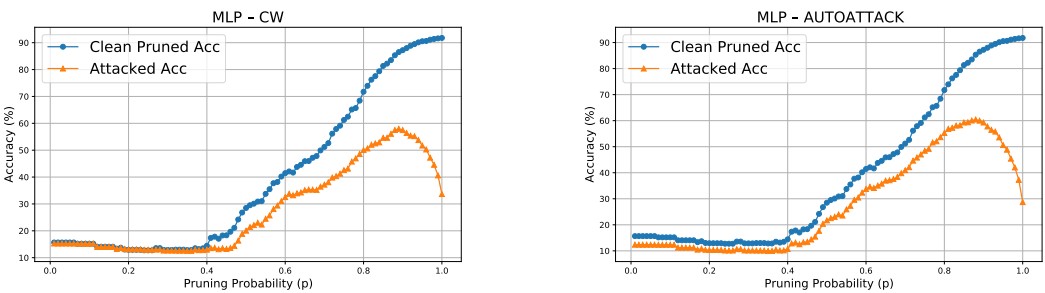

Figure 12: Clean and adversarial accuracy of MLP pruned model under the CW attack (left) and the AutoAttack (right) across varying pruning probabilities.

Figure 13 reports the results of the CW attack on the transformer (ViT), using the same experimental setup as in the previous section, while Figure 14 presents the corresponding results for AutoAttack. Consistent with earlier observations, both attacks exhibit the same characteristic behavior, further reinforcing the universality of our theoretical analysis and the inherent trade-off between pruning and optimality. In particular, we again identify a sweet spot where moderate pruning can yield gains in adversarial robustness with only a minor reduction in clean accuracy.

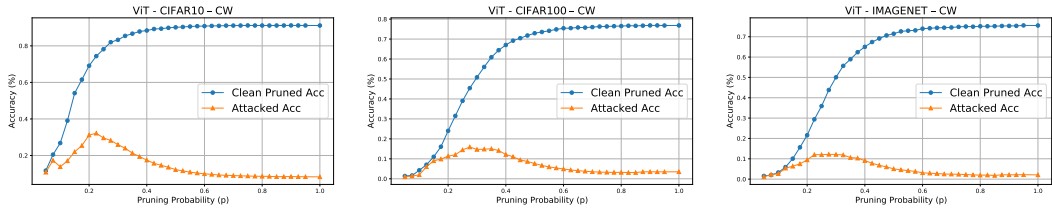

Figure 13: Clean and adversarial accuracy of pruned models under CW attack across varying pruning probabilities.

## H.6 EMPIRICAL ANALYSIS OF THE BOUNDS TIGHTNESS

In our theoretical analysis, we established a connection between the pruning probabilities and the resulting adversarial robustness of the model (Theorem 2). We now aim to empirically assess the tightness of this bound.

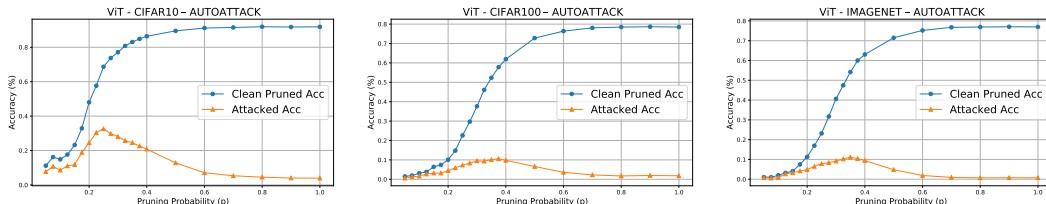

Figure 14: Clean and adversarial accuracy of pruned models under AutoAttack attack across varying pruning probabilities.

To do so, we compare the theoretical upper bound with an empirical approximation. Because computing the empirical quantity defined in Equation 1 exactly is intractable, we rely on an estimator. Specifically, for each input $x$, we sample $K = 200$ points (images) within its considered neighborhood, defined the attack budget $\epsilon$. For each sampled point, we measure the change in the model's output relative to the clean input. Figure 15 reports the results. As expected, we observe a gap between the theoretical upper bound and the empirically estimated quantity.

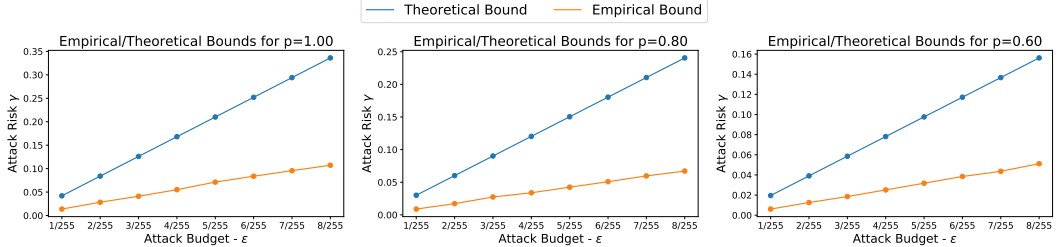

Figure 15: Empirically analyzing the tightness of the provided upper-bound, by comparing it to the Empirical estimated quantity of adversarial risk.

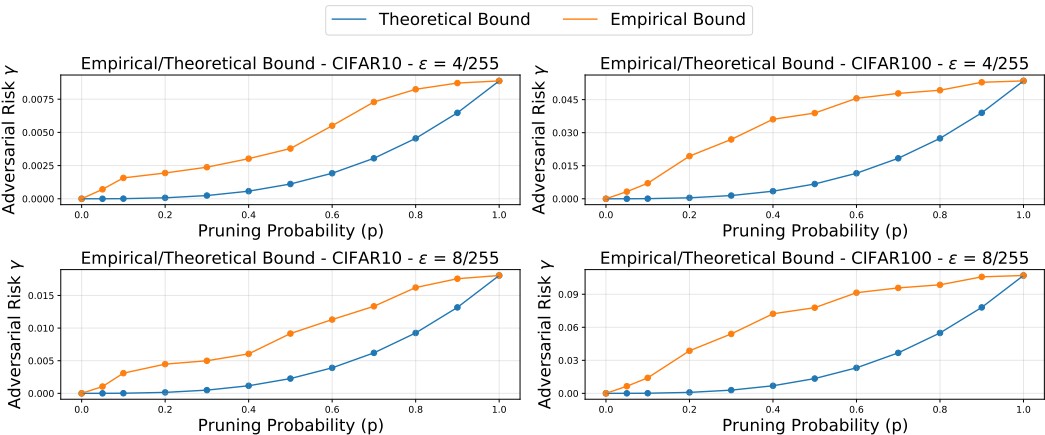

Figure 16: Empirically analyzing the tightness of the provided upper-bound in Theorem 1, by comparing it to the Empirical estimated quantity of adversarial risk.

We additionally analyze the tightness of the bounds related to the Transformer model, provided in Theorem 1. In the same manner as the previous experiment, we compute the adversarial risk through the same estimator, and we compute the theoretical bound through the provided upper-bound. Figure 16 provides the result of the analysis. As expected, as the pruning probability $p$ decreases (resulting in more pruning of the model), then resulting adversarial robustness $\gamma'$ of the model decreases reflecting enhanced robustness.

## I IMPLEMENTATION DETAILS

We start by noting that the necessary code to reproduce the results is provided in the supplementary materials and shall be made public upon publication. In what follows, we provide experimental details and hyper-parameters choices.

**MLP.** The first model considered in our study and in line with our theoretical analysis, is the Multi-Layer perceptron (MLP). Specifically, for the MNIST dataset, we used a 2-Layers MLP model, while for the CIFAR-10 dataset, we had to adapt to a 4-Layers MLP model with hidden dimensions of (8192, 4096, 2048 and 1024) to reach a satisfactory initial clean accuracy. For MNIST dataset, a satisfactory accuracy can be reached in 30 epochs, while for the CIFAR-10, we used a 100 training epochs.

**CNN.** The second model to be used was a CNN, where we considered a 4-Layers CNN model for both the CIFAR-10 and CIFAR-100. We have trained the model for 100 epochs to reach convergence.

**ViT.** For our third model, we consider a Vision Transformer. Specifically we used a Tiny ViT, which is composed of $5M$ parameters. The model is pre-trained on the ImageNet Dataset. We used the checkpoint provided by the Timm and is publicly available in HuggingFace. For all the results, we finetuned the model for 10 epochs, which was enough to reach the convergence and a satisfactory clean accuracy performance.

**Training.** For the CNN and NLP, all the experiments have been trained using the Adam optimized (Kingma & Ba, 2014) with a learning rate of $1e - 03$. For the ViT, we have used the AdamW, with a learning rate of $5e - 04$. We note that all the models have been trained without any adversarial training or other robustness-enhancing fine-tuning procedures.

**Adversarial attacks.** For the PGD and FGSM attack, we consider $\epsilon = 4/255$ (we additionally consider $\epsilon = 8/255$ for the FGSM to showcase the generality of the results). For the PDG attack, we set the number of iteration to 5. For the AutoAttack baseline, we rely on its default untargeted $\ell_\infty$ configuration as implemented in the *torchattacks* library (which we have used directly in the script without any changes), using the same perturbation budget $\epsilon = 4/255$. For the CW attack, we adopt the iterative $\ell_\infty$-constrained variant, with step size $\alpha = 2/255$ and we set the number of iterations (ascent) to 10. We additionally set the confidence margin parameter to $k = 50$. All the experiments were run using a single NVIDIA L4 GPU and took around 200 GPU hours to obtain all results.

**On the Score-Based pruning.** The main assumption in score-based pruning based on gradient is that the weights contributing little to the loss should have minimal impact if removed, and therefore can be pruned first. Formally, the strategy is mainly based on the Taylor approximation of the loss with respect to each weight is used. Specifically, for every weight $W$, the model is evaluated on several batches to compute the cross-entropy loss $\mathcal{L}$, after which backpropagation provides the corresponding gradient $\nabla_W \mathcal{L}$. Consequently, for each batch, we compute an element-wise scoring through a Hadamard product $| W \odot \nabla_W L |$. These scores are accumulated and averaged across a fixed number of batches, yielding for each parameter $w_{ij}$ a final importance measure, which can be formulated as:

$$s_{ij} = \mathbb{E}_{\text{batch}}\Big[ \, | \, w_{ij} \frac{\partial L}{\partial w_{ij}} \, | \, \Big],$$

This quantity is exactly an estimator of the first-order change in loss resulting from removing that weight. Based on the produced score, the pruning is applied by removing the fraction of weights with the smallest scores. Empirically, we use 20 batch to construct the estimator of the gradient.

**On the Magnitude-based pruning.** In addition to score (gradient) based pruning, we also consider standard magnitude pruning, where the importance of each parameter is measured solely by the absolute value of its weight. We follow the same setting as the one provided in the Preliminaries (Section 3).

