# OpenReview forum: "When Less Is More: Uncovering the Robustness Advantage of Model Pruning"
_ICLR.cc/2026/Conference — Submitted to ICLR 2026_

### Official Review · Reviewer_9Euk · 2025-10-26

**Soundness:** 3
**Presentation:** 2
**Contribution:** 3
**Rating:** 6
**Confidence:** 4

**Summary:**

This paper explores the relationship between model pruning and adversarial robustness by examining how different pruning strategies and parameters influence adversarial behavior. The theoretical analysis reveals that pruning has a positive effect on adversarial risk while negatively affecting natural performance. This dual impact clarifies the trade-off between robustness and accuracy, offering a principled foundation for understanding adversarially robust pruning.

**Strengths:**

**High readability with clear logical flow**

The paper is logically well-structured and easy to follow. The preliminaries section clearly defines the concepts of adversarial robustness and model pruning. Each step in the theoretical analysis builds coherently on the previous ones, maintaining a strong logical flow that enhances readability and overall comprehension.

**Insightful analysis of adversarial risk and its relationship to natural performance**

The analysis effectively highlights how pruning simultaneously reduces adversarial risk and degrades natural performance. This insight helps explain both the regularization benefits of mild pruning and the challenges of maintaining robustness under aggressive compression.

**Weaknesses:**

**W1: Adversarial risk does not directly reflect model robustness**

Although Theorems 1 and 2 indicate that a higher pruning probability leads to a smaller adversarial risk, this only measures the distance between predictions on clean and perturbed inputs, $x$ and $\hat{x}$. In practice, a higher pruning ratio often causes a significant drop in natural accuracy. Therefore, despite the reduced adversarial risk, the absolute robustness of the pruned model may still decline. Consequently, it may be inaccurate to directly claim that pruning has a positive impact on adversarial robustness (see lines 288–291).

**W2: Limited discussion on layer-wise pruning ratios**

As shown in [1], effective pruning should account not only for the importance of individual parameters but also for layer-wise pruning ratios to better preserve both adversarial robustness and natural performance. Given that adversarial risk is bounded by natural performance, how can the proposed analysis in Proposition 1 be extended to interpret the variance of pruning ratios across layers?

**W3: Limited evaluation on advanced adversarial perturbations**

The evaluation primarily relies on PGD and FGSM attacks, which do not fully capture the model’s robustness. A more comprehensive evaluation should include stronger and more diverse attacks, such as C&W [4], AutoPGD, and AutoAttack [5].

---
### References

[1] Zhao and Wressnegger, "Holistic Adversarially Robust Pruning," ICLR 2023.
[2] Sehweg et al., "Hydra: Pruning Adversarially Robust Neural Networks," NeurIPS 2020.
[3] Ye et al., "Adversarial robustness vs. model compression, or both?" ICCV 2019.
[4] Carlini et al., "Towards Evaluating the Robustness of Neural Networks," IEEE S&P 2017.
[5] Groce and Hein, "Reliable Evaluation of Adversarial Robustness with an Ensemble of Diverse Parameter-free Attacks," ICML 2020.

**Questions:**

**Q1: How can the proposed magnitude-based pruning analysis be extended to structured pruning?**

The analysis in Section 3 is based on weight pruning. How might these findings generalize to structured pruning? Does the relationship between pruning probability and adversarial risk still hold in that context?

**Q2: Does the theoretical relationship persist after post-pruning fine-tuning?**

Previous works on adversarially robust pruning [1, 2, 3] include adversarial fine-tuning to recover performance. Since fine-tuning updates model weights, does the established relationship between pruning ratio and adversarial risk remain valid after fine-tuning?

**Q3: Can pruning improve adversarial robustness before harming natural performance?**

Empirical studies [1, 3] suggest that mild pruning can enhance robustness through regularization effects. As pruning progresses from minor to aggressive levels, does robustness first improve before degradation in natural accuracy occurs? Is it possible to derive an optimal pruning strategy that maximizes adversarial robustness without compromising natural performance?

---

> ### Author Response · Authors · 2025-11-20
>
> We appreciate the reviewer's detailed feedback, which has been instrumental in improving our work. Several raised points highlighted areas where additional clarity was needed, and we have worked to address these in the revised manuscript. Below, we provide responses to the specific concerns along with corresponding revisions.
>
> **W1 & Q3-Adversarial risk, Robustness and Performance:** We appreciate the opportunity to clarify this important distinction. The adversarial risk metric controls the consistency of model outputs within the perturbation neighborhood, ensuring predictable behavior for perturbed inputs. However, as the reviewer correctly identifies, this metric alone does not guarantee meaningful performance. A trivial constant function would achieve zero adversarial risk but would be completely uninformative, illustrating why optimizing robustness in isolation can lead to degenerate solutions.
>
> This motivates the second component of our analysis: optimality bounds that measure how closely the pruned model's outputs match those of the original model. By combining both robustness and optimality bounds, we formulate the problem as a bi-objective optimization (Section 5.3) and provide an algorithm to identify optimal pruning probabilities that balance both objectives. Section 6.4 validates this approach empirically.
>
> Beyond parameter selection, our framework addresses a fundamental question: can we improve robustness through pruning without sacrificing performance? Our theoretical analysis establishes the existence of a Pareto front between these objectives (Lemma 2, Appendices F and G), revealing an inherent inverse relationship. This demonstrates that maintaining the exact performance of the unpruned model while gaining robustness is theoretically infeasible. Our framework therefore provides a principled approach to navigate this trade-off, allowing practitioners to make informed decisions about the acceptable level of performance degradation for desired robustness improvements.
>
> **W2-Regarding layer-wise pruning:** We appreciate this insightful observation. Our theoretical analysis indeed allows for layer-specific pruning ratios (denoted as $p_{i,j}^{(\ell)}$​), though our initial experiments used uniform pruning across layers. Following the reviewer's suggestion, we have now leveraged our optimization framework and accompanying algorithm to investigate layer-wise pruning strategies empirically. The results confirm the reviewer's intuition. Figure 5 compares uniform versus layer-wise pruning, revealing that the optimization naturally tends to prune later layers more aggressively. This strategy yields improved trade-offs, demonstrating the practical value of our flexible theoretical framework.
>
> **W3-On the considered Attacks:** While our theoretical analysis does not assume any specific adversarial attack, our initial empirical validation focused on FGSM and PGD. Following the reviewer's recommendation, we have now extended our evaluation to include the CW attack and AutoAttack. The results are provided in Appendix H.5 of the updated manuscript and exhibit consistent patterns across all attack methods, confirming the generalization of our findings beyond the originally tested attacks. We thank the reviewer for suggesting this extension.
>
> **Q1-Regarding Structure Pruning:** We appreciate this insightful suggestion. In our current manuscript, we focus on unstructured pruning, modeling pruning parameters as probabilities $p_{i,j}^{(\ell)}$​ for individual weights. Structured pruning can be naturally incorporated within this framework as a special case where probabilities are applied at the group level rather than to individual weights. Specifically, this amounts to tying probabilities across predefined weight groups (neurons, channels, or filters ..). Formally, for a group $G_k^{(\ell)}$ in layer $\ell$, all weights within the group share a common pruning probability​. Under this formulation, our robustness bounds can be directly adapted using group-level analysis. Consequently, both the robustness and optimality bounds, along with the formulated bi-objective optimization problem, extend naturally to structured pruning with appropriate modifications to the solver algorithm.
>
>
> **Q2-On Post-Pruning fine-tuning:** In our current framework, we consider post-pruning without fine-tuning, where the pruning probabilities $p$ are directly connected to the original model weights. Fine-tuning after pruning introduces weight updates that break this direct connection, making it challenging to apply our current theoretical analysis. The evolved weights would require re-deriving bounds that account for the optimization trajectory during fine-tuning.
>
> Nonetheless, this is a promising direction for future work. Our framework could potentially inform fine-tuning strategies that simultaneously optimize both performance recovery and robustness enhancement, leading to more principled approaches than current empirical methods.

---

> > ### Comment · Reviewer_9Euk · 2025-11-26
> > **Response to the authors**
> >
> > I appreciate the authors' effort in the rebuttal. Most of my concerns have been addressed. However, two questions remain unclear and appear not fully aligned with prior work, particularly regarding CNN models with higher structural complexity.
> >
> > **Gaining robustness while preserving natural performance is infeasible**
> >
> > As shown in prior work [A,B], moderate pruning can improve the adversarial robustness without negatively impacting natural accuracy. However, the experimental setup differs. In this submission, MLP and CNN models are structurally simpler than models such as ResNet or VGG commonly used in related work. It is reasonable to expect that larger models have higher intrinsic structural redundancy, which potentially allows pruning to improve the robustness and preserve the natural accuracy at the same time.
> >
> > How do the authors account for the model's intrinsic redundancy in the proposed theoretical findings, or are they limited in the case where the baseline model needs to be already compact?
> >
> > [A] Xiao et al., "Training for Faster Adversarial Robustness Verification via Inducing ReLU Stability," ICLR 2019.
> > [B] Guo et al., "Sparse DNNs with Improved Adversarial Robustness," NeurIPS 2018.
> >
> > **The optimization is prone to pruning later layers**
> >
> > What do the layer-wise pruning probabilities look like for models with higher structural complexity, e.g., ResNet18 or VGG16?

---

> > > ### Author Response · Authors · 2025-11-26
> > >
> > > We thank the reviewer for the continued engagement and the thoughtful follow-up questions. We are glad that the additional experimental results and theoretical clarifications addressed the majority of concerns. Below, we address the remaining points.
> > >
> > > **Q1 - On the Clean/Attacked Robustness trade-off:** We appreciate the opportunity to clarify how our theoretical framework relates to the empirical findings in [A, B].
> > >
> > > Our analysis characterizes clean performance through the optimality quantity $\zeta$, which measures how far the pruned model's outputs deviate from those of the original model. Crucially, $\zeta$ is not a direct proxy for accuracy: its impact on clean accuracy depends on the geometry of the output manifold. Models with higher intrinsic redundancy can tolerate larger $\zeta$ deviations with minimal changes in accuracy, while compact models with sensitive decision boundaries will exhibit stronger degradation for the same $\zeta$.
> > >
> > > This relationship is reflected in our experiments. Figure 1 shows that $\gamma$ (robustness) and $\zeta$ (optimality) exhibit an inverse dependence on the pruning probability $p$. Figures 2 and 3 then illustrate how these theoretical quantities translate into clean and attacked accuracy: clean accuracy remains nearly unchanged over a range of moderate pruning levels, particularly for ViT and deeper MLPs, depicted in Figure 4, while adversarial accuracy increases as $\gamma$ decreases.
> > >
> > > This is precisely why the "free robustness" phenomenon reported in [A, B] is consistent with, and in fact predicted by, our framework. The Pareto front characterizes the fundamental trade-off between $\gamma$ and $\zeta$, but the translation from $\zeta$ to actual accuracy loss depends on model capacity. In redundant architectures, there exists a regime where $\zeta$ increases (due to pruning) but accuracy remains essentially unchanged, yielding the "free lunch" observed empirically in prior work. The Pareto front in Figure 6 makes this correspondence especially apparent: in the MLP/MNIST setting, adversarial accuracy increases from ~38% to ~50% while clean accuracy changes by less than 0.5%.
> > >
> > > Regarding the reviewer's point on larger models, we fully agree that architectures with higher intrinsic redundancy should exhibit an even wider region where pruning improves robustness without meaningful accuracy loss. Our experiments confirm this scaling effect (beyond deeper MLPs in Figure 4): the Base ViT, which is substantially larger and more complex than ResNet-50 in terms of parameters, displays the same trend observed in Figure 2 of [B], where considerably more pruning is required before clean accuracy begins to degrade. This is fully consistent with the reviewer's intuition and with prior empirical results on large CNNs.
> > >
> > > In short, our theoretical trade-off between $\gamma$ and $\zeta$ does not contradict prior findings. Rather, it provides the formal grounding that explains *why* larger architectures exhibit a broader pruning regime in which robustness improves while clean accuracy remains essentially unchanged. The key insight here is that the Pareto trade-off between $\gamma$ and $\zeta$ translates to clean and attacked accuracy through the lens of model complexity and output manifold geometry. As a result, redundant architectures can absorb larger $\zeta$ deviations with minimal accuracy loss (within 1-2%), which aligns with the phenomena reported in prior work and is now explained by our framework.
> > >
> > > **Q2 - On layer-wise pruning in deeper architectures:** As noted in our previous response, the trade-off formulation requires tractable Lipschitz-bound expressions, which we have established for MLPs, shallow CNNs with comparable Lipschitz behavior, and Transformers. The ViT experiments, in particular, demonstrate applicability to architectures with considerable depth and complexity. Extending the formal trade-off characterization to ResNets or VGG-style networks would require handling residual connections and normalization layers, which interact non-trivially with the $\gamma$-$\zeta$ decomposition. We view this as a valuable direction for future work and will update our limitations section accordingly.
> > >
> > > We hope these clarifications fully address the reviewer's remaining questions. We are grateful for the constructive feedback and the insightful directions it has highlighted.

---

> > > > ### Comment · Reviewer_9Euk · 2025-11-27
> > > >
> > > > Thank you for providing these additional clarifications. They adequately resolve my concern about the model’s structural redundancy. As my assessment is already positive, I keep my score unchanged.

---

> > > > > ### Author Response · Authors · 2025-11-27
> > > > >
> > > > > We thank the reviewer for the dedicated time in reviewing the paper and interacting with our provided rebuttal and the further insightful discussion. We are glad that we managed to answer their concerns and questions, and we additionally appreciate the provided positive score. We will carefully incorporate elements from the discussed elements into the revised paper.

---

### Official Review · Reviewer_fjkr · 2025-10-28

**Soundness:** 2
**Presentation:** 2
**Contribution:** 1
**Rating:** 2
**Confidence:** 5

**Summary:**

This paper investigates the theoretical and empirical relationship between model pruning and adversarial robustness, aiming to provide a formal theoretical framework that links pruning parameters to adversarial risk.

**Strengths:**

- The approach is interesting.
- Studies on adversarial robustness and pruning can be relevant.
- The used architectures are modern and up-to-date.

**Weaknesses:**

**Outcomes of the study and relation to prior work:** The paper starts with good premises of providing a formal definition, characterizing a trade-off, and validate insights through experiments. However, in doing so, I believe that the authors end up with some overly generic outcomes which do not add much value to our current understanding of the relationship between robustness and pruning. This is also made more severe by the lack of a critical paragraph in the manuscript which reflects and compares the provided contributions with what is already known from the state-of-the-art. Specifically:
- *Definition of adversarial robustness in the context of pruning:* I do not see the necessity of providing this definition in Equation 3 and claim it as a contribution, since it is a simple definition of certified robustness on a given bound adapted on a pruned model (which is still a model) [ext_ref_1].
- *Outcome from sections 4 and 5:* It is well-known that pruning may or may not enhance adversarial robustness [int_ref_1, ext_ref_2], but that is not being considered in the discussion and does not add much to what is known from previous work. It is also very well-known in adversarial robustness that there exist an inherent trade-off between accuracy and adversarial robustness  [ext_ref_3]. Therefore, in general, the outcome of the theorems cannot be limited to a simple observation which traces already known concepts, but it is supposed to provide something new or at least some new and well-supported opinion highlighted by the theorem itself or further experiments.
- *Experimental validation:* What one infers from the experiments is that there is an initial improvement given by pruning, which is then reduced as sparsity increases. Once again, prior work has largely discussed the regularization effect of pruning on adversarial robustness [int_ref_1, ext_ref_2].

**Experiments:**
- *Adversarial robustness evaluation:* In adversarial robustness, it is now well-known that using attacks such as FGSM or PGD can lead to overestimation. Hence, the community has moved a step forward from relying solely on these attacks [ext_ref_4], and advocated for more reliable approaches such as the AutoAttack framework [ext_ref_5]. In addition, these advances have also been put forward in adversarial pruning literature [ext_ref_2], but this is not being considered in the work. I believe that the choice of relying on these attack approaches makes the manuscript largely behind already well-known advances in adversarial robustness literature.
- *Missing details:* It is not clear from the manuscript how are scores computed in the score-based approach used in 6.3.2, as I not find it specified in the setup description. I also fail in understanding whether the pre-trained models are robust against adversarial attacks, and if yes how they have been adversarially trained. In addition, I read in the appendix that the used perturbation bound is $\epsilon=4/255$, but it is common practice to choose the perturbation bound based on the used dataset (e.g., $\epsilon=8/255$ for CIFAR-10 and $4/255$ for ImageNet).

**Other comments and suggestions, not necessarily minor:**
- Please write a paragraph in the related work section describing how the contributions and outcomes of the theorems differ from known concepts, and additionally consider important missing citations on highly related work such as [ext_ref_2] (which provides a taxonomy and discusses the work and advances in the literature) or other robustness/pruning related papers. I also suggest to follow the adversarial robustness guidelines provided there.
- Increase legend fontsize in Figure 2, as they are too small and not easily readable.
- I found abstracts and introduction slightly misaligned. I usually find easier to read papers where what is said to be done in the abstract is then extended and reflected in the introduction. For instance, in the abstract the authors start by saying that "We first show that the pruning strategy and associated parameters play a critical role in determining the robustness of the resulting pruned model", but this looks to me more of a contribution that the authors intend to provide after the theorems and discussion. But clearly, this is more of a subjective preference and for sure not a limitation.
- The authors mention that users can then use their formulated trade-off to find optimal strategies, but it is not clear to me how one can choose the strategy based on the provided formalization.

[ext_ref_1]: Hein, Matthias, and Maksym Andriushchenko. "Formal guarantees on the robustness of a classifier against adversarial manipulation." Advances in neural information processing systems 30 (2017).

[int_ref_1]: Artur Jordao and Hélio Pedrini. On the effect of pruning on adversarial robustness. In Proceedings of
the IEEE/CVF International Conference on Computer Vision, pp. 1–11, 2021.

[ext_ref_2]: Piras, Giorgio, et al. "Adversarial pruning: A survey and benchmark of pruning methods for adversarial robustness." Pattern Recognition (2025): 111788.

[ext_ref_3]: Zhang, Hongyang, et al. "Theoretically principled trade-off between robustness and accuracy." International conference on machine learning. PMLR, 2019.

[ext_ref_4]: Carlini, Nicholas, et al. "On evaluating adversarial robustness." arXiv preprint arXiv:1902.06705 (2019).

[ext_ref_5]: Croce, Francesco, and Matthias Hein. "Reliable evaluation of adversarial robustness with an ensemble of diverse parameter-free attacks." International conference on machine learning. PMLR, 2020.

**Questions:**

- What is the added value of formally defining adversarial robustness for pruned models?
- How does the outcome of Sections 4 and 5 differ from what is already known from the state-of-the-art? Could you explain what new theoretical insight these results provide beyond re-stating known effects, and how they relate to known literature?
- Why did you choose to evaluate robustness only with FGSM or PGD attacks, given that these are known to overestimate robustness?
- How are the scores computed in the score-based pruning method described in Section 6.3.2?
- Were the models pre-trained with adversarial training, or are they standard models pruned? If they were adversarially trained, please clarify which training method and perturbation bounds were used.
- Why is the perturbation bound fixed to $\epsilon=4/255$ for all datasets?
- The paper mentions that the proposed trade-off formulation can help users identify optimal pruning strategies. Could you elaborate on how a practitioner should concretely use this trade-off in practice?

---

> ### Author Response · Authors · 2025-11-20
>
> We thank the reviewer for the thorough feedback and review, and for all the pointed out elements that helped enhance our manuscript. Following the different raised elements, we have updated the manuscript, and in what follows we aim to respond to the raised questions and weakness and also pointing out to the relevant edits that were made:
>
> **W1/Q1 - Regarding the definition:** We apologize if we were giving the impression that the provided definition is a contribution. It is a simple adaptation of previous work (such as the one pointed out by the reviewer and others [1, 2] to the context of pruning. We believe that the two  formulations in Definition 1 (resp. Definition 2) were important to be able to put some formalism on the concept of adversarial robustness (resp. optimality), and consequently be able to provide the theoretical derived upper-bounds in the different theorems and propositions. We have added these references to the manuscript (Section 4.1) to clarify that this is not a contribution but rather an adaptation of previous work.
>
> —
>
> [1] Robustness between the worst and average case. - Rice & Al. - NeurIPS 2021
>
> [2] Theoretical evidence for adversarial robustness through randomization. Pinot & Al. NeurIPS 2019.
>
>
> **Q2/W2 - On the added value of the analysis and its link to previous work:** We apologize if this wasn’t expanded enough in our original manuscript. Typically, previous work has shown the existence of the trade-off robustness/optimality in the case of pruning empirically but without specifying how we could optimally use the best-parameters (which in our opinion can be of great advantage to the final user). In our work, we have taken a step further by trying to theoretically understand this effect and its link to the pruning parameters/probabilities. By deriving the robustness upper-bound and the optimality upper-bound, we are able to formalize this trade-off mathematically as a bi-objective optimization problem (detailed in Section 5.3) and results in two main elements for the final user:
> - We provide an adaptation of the Coordinate-descent algorithm to solve this problem (Detailed in Appendix E) and get directly the optimal pruning parameters to be used depending on the considered context, which was also empirically validated in Section 6.5.
> - Using this trade-off, we can theoretically analyze the existence of a sweet-spot in the specific context of pruning (which was also empirically analyzed in the paper [ext_ref_3] referenced by the reviewer); where a pareto-trade off can be observed (Appendix F and G).
>
> In response to the reviewer’s suggestion, we added some elements in the Related Work section but also a complete discussion in Section 7 that explicitly highlights the subsequent research directions our work enables over prior studies.
>
> **Q3/W3 - Experimental Validation:** As identified by the reviewer, in the empirical analysis we have indeed only focused on FGSM and PGD (as we thought that is enough to illustrate the worth of the study). Consequently and in line with the reviewer’s proposition, we have now extended the results to take into account the CW Attack and the AutoAttack. The results are provided in the updated manuscript in Appendix H.5 and showcase the same patterns, confirming therefore the generalization of the results. We thank the reviewer for pointing out this extension.
>
> **Q4 - On the Score-Based Pruning:** For this line of pruning, we simply compute the scores $s_{i,j}^{(\ell)}$ by multiplying the weight itself with the gradient quantity ($\nabla_W \mathcal{L}$). This is exactly an estimator of the first-order change in loss resulting from removing that specific weight. We have extended details about this in Appendix I
>
> **Q5/Q6 - Regarding some implementation details:** We have added some details in the Appendix regarding all the elements. Specifically, we would like to note that in the current context, the models that were used are not trained using adversarial training but rather normal training. Nonetheless, our theoretical study doesn’t assume any information about the model. So after adversarial training, we can directly consider the weights of the resulting model, and then apply our proposed trade-off optimization objectives. Additionally , for the original manuscript, we have chosen to operate through an $\epsilon=4/255$, there was no specific reason for such choice. We don’t see how such this choice can affect the results and the existence of the trade-off, and therefore we have provided results for $\epsilon=8/255$ in Appendix H.4.

---

> ### Comment · Reviewer_fjkr · 2025-11-25
> **Response to the authors.**
>
> Thank you to the authors for the thorough and precise rebuttal.
>
> **W1/Q1 - Needed formalism:** I understand the need to introduce formalism, and I am happy that the authors re-calibrated that as an adaptation of existing formulation, since I think it was a bit misleading.
>
>
> **W2/Q2 - Added value:** I understand, but I still struggle to make this practically meaningful. Can the authors elaborate more on that?
>
> **W3/Q3 - AutoAttack:** I appreciate the experimental effort in this specific point.
>
> **Q4 - Score-based Pruning:** Please be aware that score-based pruning methods are normally conceived differently in the relevant literature. For instance, methods such as HYDRA and HARP are considered score-based, as they optimize scores with respect to adversarial robustness and then choose the corresponding mask. I would frame this as Taylor pruning, which is similar to the approach used by LeCun in Optimal Brain Damage. While writing this, I also realized that the paper somehow disregards previous efforts made in the state-of-the-art of pruning methods for robustness, and limits the analysis to magnitude-based and Taylor-based methods.
>
> **Q5/Q6 - Implementation details:** I believe that choosing different norms than those used by the state-of-the-art limits the relevance/comparability of the outcome from this analysis. This relates to my previous concern of relevance in W2/Q2 and practical applicability in Q7.
>
> At this stage, I am keeping my score. I am happy to further discuss this.

---

> > ### Author Response · Authors · 2025-11-25
> >
> > We thank the reviewer for the thoughtful engagement throughout this discussion and for the detailed analysis of our submission. We are glad that the additional experimental results and theoretical clarifications were found useful, and we address the remaining points below.
> >
> > **Regarding our work:** Prior work observed empirically that pruning can improve adversarial robustness, but stopped at this observation without providing a principled method to exploit it. In practice, the only option available to practitioners was exhaustive search over pruning rates.
> > Our work bridges this gap by transforming this empirical observation into a practically actionable tool as follows:
> >
> > - *Theoretical foundation:* Under mild tractability assumptions, we derive bounds explicitly linking pruning parameters $p_{i,j}^{\ell}$ to both robustness and optimality (clean accuracy), moving beyond intuition to provide quantitative relationships
> >
> > - *Formal trade-off characterization:* From these bounds, we prove that robustness and optimality form a Pareto front, mathematically confirming the empirically observed trade-off and establishing that robustness gains necessarily come with an accuracy cost
> >
> > - *Practical algorithm:* Using this formulation, we propose a coordinate-descent algorithm that automatically solves the resulting optimization problem. Concretely, a user specifies an acceptable accuracy drop (e.g., $\leq 2$%), inputs their trained model, and our algorithm outputs layer-wise pruning probabilities that maximize robustness within that constraint, hence replacing exhaustive grid search with principled optimization.
> >
> > As such, our work advances the field from observing that pruning enhances robustness to providing a principled method that identifies optimal pruning configurations for any given performance tolerance.
> > We thank the reviewer for emphasizing this key aspect of our work and and remain happy to provide any further clarification.
> >
> > **On the magnitude/scoring pruning:** We thank the reviewer for this clarification. Our use of "score-based pruning" referred to the fact that the method derives a scoring matrix $s$ from gradient information (via Taylor expansion). We agree this terminology is overloaded and potentially confusing; we will revise the manuscript to explicitly frame our method as Taylor-based pruning with appropriate references.
> >
> > Regarding related work, we note that Section 2 (starting at line 94) discusses pruning methods related to adversarial robustness. As stated in the introduction, our analysis focuses on post-training pruning at inference time without fine-tuning, which is why we restricted comparisons to norm-based and Taylor-based techniques. Methods like HYDRA or HARP, which optimize robustness-oriented scores during training and require retraining, fall outside our scope.
> > That said, our framework could naturally extend to such methods: after their scoring procedure produces a matrix $s$, one could apply our analysis to derive optimal pruning masks.
> >
> > We will clarify this distinction in the limitations section and highlight this as a promising direction for future work.
> >
> > **Regarding the attack budgets:** We thank the reviewer for emphasizing the importance of standard evaluation protocols. Following this suggestion, we have included results using $\epsilon = 8 / 255$  for both CIFAR-10 and CIFAR-100 in the Appendix, ensuring alignment with state-of-the-art benchmarks.
> >
> >
> > We hope these clarifications address the reviewer's remaining concerns. We remain happy to continue this constructive discussion and to provide any further details that would be helpful.

---

> > > ### Author Response · Authors · 2025-11-28
> > >
> > > Dear Reviewer,
> > >
> > > Thank you once again for your valuable time and insightful feedback, which provided valuable guidance for refining our work and enhance the quality of our manuscript. We are also grateful for your engagement in the discussion and interacting with our rebuttal. We have tried to answer the raised ambiguities in our previous response and as the discussion period reaches its final part, we would be grateful to know whether our latest responses have resolved your concerns and raised points or if any questions remain. We are happy to provide further clarification as needed.
> > >
> > > Thanks,
> > > Authors

---

> ### Comment · Reviewer_fjkr · 2025-11-28
> **Response to authors**
>
> Dear authors, thanks, I read your response to my comment.
> You are also right in demanding a response, and I apologize if this last one took me some days.
>
> My concerns regarding practicality/relevance/impact of the derived insights partially remain, and I think that the first submission had some oversights, more or less marginal, which I should still take into account in the final evaluation.
>
> However, besides appreciating your rebuttal spirit, I am now starting to appreciate also the contribution overall. I am raising my score from 2 to 4 because that best expresses my feelings about your paper now.
> I am then looking forward to the internal discussion/decision time: if some fellow reviewer hardly feels that this paper requires acceptance, I will be happy to discuss and open to be “nudged” toward acceptance, based on actual facts of course.
>
> Thank you, you did a good job with both paper and rebuttal.
>
> PS: I will increase the score as soon as the edit option, following privacy violations, will be up again.

---

> > ### Author Response · Authors · 2025-11-28
> >
> > Dear Reviewer,
> >
> > We thank you for your thorough engagement throughout this discussion which has been very valuable in improving our manuscript. Should any further ambiguities need clarification, we are happy to address them.
> >
> > Thanks,
> > Authors

---

### Official Review · Reviewer_BmZv · 2025-10-31

**Soundness:** 3
**Presentation:** 3
**Contribution:** 3
**Rating:** 6
**Confidence:** 3

**Summary:**

The paper explores how model pruning influences adversarial robustness. It proposes a formal theoretical framework that relates pruning probabilities to adversarial risk bounds, then validates the predictions experimentally on standard architectures and datasets. The work finds that pruning can improve robustness but introduces a clear trade-off with clean accuracy. While the theoretical part is mathematically sound, some assumptions (e.g., Lipschitz continuity, uniform pruning) may limit applicability.

**Strengths:**

1. The paper provides the first theoretical understanding of the impact of pruning on adversarial robustness, establishes a unified probabilistic pruning framework, and extends beyond existing empirical studies. The theoretical framework is clearly articulated and builds upon solid mathematical foundations.

2. The study contributes to an underexplored intersection between compression and robustness.

3. The experiment covered various models, datasets, and attacks, verifying the generalization ability of the theory.

4. The paper reveals the implicit regularization effect of pruning, providing a new perspective for robust model design.

**Weaknesses:**

1. The theoretical analysis assumes uniform pruning probabilities and Lipschitz-continuous layers, which may not generalize to real-world architectures.
2. There is little discussion of computational efficiency or practical trade-offs.
3. Focus is placed solely on magnitude-based and score-based pruning, and the exclusion of other strategies (e.g., structured pruning) could restrict the applicability of the findings.
4. The experimental scope employs only FGSM and PGD attacks; the lack of evaluation against more sophisticated attacks (such as adaptive attacks) means the robustness assessment is not comprehensive.

**Questions:**

1. Theoretical compactness: Is the derived upper bound compact in practice? Are there any experimental data that can be directly compared with the upper bound?
2. Does the analysis extend to stochastic or dynamic pruning applied during training?
3. What is the intuition behind using Frobenius norms in Theorem 2 rather than spectral norms?
4. How were the pruning rates selected for each architecture, and do they differ by dataset?
5. Is the computational cost (such as fraction calculation) during the pruning process significant?

---

> ### Author Response · Authors · 2025-11-20
>
> We thank the reviewer for the insightful review and constructive critique. The recognition of our theoretical framework is much appreciated. We have carefully considered the points raised regarding clarity and have revised the manuscript accordingly. What follows addresses each concern and details the changes implemented.
>
> **W1 - On the considered assumptions :** We appreciate the opportunity to clarify these aspects of our framework. In our main theoretical analysis (beyond Theorem 1), pruning is not assumed to be uniform. Rather, we use element-specific notation $p_{i,j}^{(\ell)}$ for each component within layer $\ell$, allowing heterogeneous pruning across the network. For Theorem 1, we adopted uniform pruning notation to simplify the bound presentation, though extension to element-specific pruning is straightforward.
>
> Regarding activation functions, our framework only requires them to be $1$-Lipschitz, which is satisfied by the majority of commonly used activations in practice. We have revised the problem setup in the manuscript to make these assumptions more explicit and thank the reviewer for highlighting this point.
>
> **W2 - On the practicality of the trade-off:** We appreciate this feedback and have substantially expanded our treatment of this topic. In the revised manuscript, Section 5.3 now formalizes the trade-off as a bi-objective optimization problem. We provide an adapted Coordinate Descent algorithm to solve it (detailed in Appendix E) and validate the approach experimentally in Section 6.4. Additionally, we have derived sufficient conditions for the existence of this trade-off through a Pareto front analysis (Appendices F and G).
>
> **W4 - Regarding the considered attacks:** We appreciate this suggestion. In our original manuscript, we focused on FGSM and PGD to illustrate the existence of the trade-off. Following the reviewer's recommendation, we have now extended our evaluation to include the CW Attack and AutoAttack. The results are provided in the updated manuscript (Appendix H.5) and exhibit the same patterns, confirming that our findings generalize across different attack methods. We thank the reviewer for suggesting this valuable extension.
>
> **Q1 - On the compactness of the upper-bounds**: We thank the reviewer for this valuable suggestion. In our original manuscript, we empirically validated that adversarial risk decreases with the pruning value, as depicted in Figure 1. We agree that a thorough study of bound tightness is important, and we have now provided this analysis in Appendix H.6 of the revised manuscript. Specifically, we compute both the empirical risk (using a sampling estimator within the perturbation neighborhood) and the corresponding theoretical upper bound to assess tightness for both Transformer-based models and MLPs.
>
> **Q2 - On the extension to dynamic pruning during training:** We appreciate this suggestion. In the current manuscript, we focus on post-training pruning. Extending our framework to dynamic pruning during training is indeed a promising direction. We envision two potential approaches: first, our derived upper bounds could serve as additional constraints in the training objective, enabling controlled robustness throughout the learning process. Second, the bi-objective optimization could be solved periodically (e.g., after each epoch or every $k$ epochs) to adaptively update pruning values before continuing training. We believe both directions warrant future investigation.
>
> **Q3 - On the usage of Frobenius norm:** We appreciate this question. The Frobenius norm is essential for our theoretical derivations, as key steps in our analysis rely on its specific properties and do not readily extend to other norms. Generalizing to alternative norms would require additional assumptions, such as specific constraints on the distribution of model weights. We believe such assumptions would be overly restrictive and limit the practical applicability of our results. Our choice of norm allows us to maintain generality while providing meaningful theoretical guarantees.
>
> **Q4 - On the choice of pruning rates:** For the original experiments, we have tried a range of pruning probabilities between [0,1] to make sure we showcase the existence of the trade-off. This has been extended in the case of the trade-off, where the optimizer directly finds the optimal pruning values for the considered dataset.
>
> **Q5 - On the computation cost:** We appreciate this practical consideration. The computational overhead depends on the chosen post-pruning strategy. In practice, both the norm-based and score-based pruning methods considered in our work are computationally efficient, typically requiring only a few seconds to execute. For the optimization problem, our experiments show that 20-40 iterations are generally sufficient to achieve convergence. We have added additional details regarding computational complexity of the Algorithm in Appendix E.

---

> > ### Author Response · Authors · 2025-11-28
> >
> > Dear Reviewer,
> >
> > Thank you once again for your valuable time and insightful feedback, which provided valuable guidance for refining our work. We are also particularly grateful for acknowledging the worth our theoretical and empirical insights, and for the positive score. As the discussion period reaches its final part, we would be grateful to know whether our latest responses have resolved your concerns and raised points or if any questions remain. We are happy to provide further clarification as needed.
> >
> > Thanks,
> > Authors

---

### Official Review · Reviewer_w3kp · 2025-10-31

**Soundness:** 3
**Presentation:** 3
**Contribution:** 3
**Rating:** 6
**Confidence:** 3

**Summary:**

This paper provides a theoretical analysis that formalizes the impact of pruning on the adversarial risk of DNNs (Transformer-based Models and MLPs). Specifically, this paper derives upper bounds for the adversarial risk of pruned models and for the performance deviation (optimality) of the pruned model from the original. These bounds are used to characterize a trade-off between adversarial robustness and clean accuracy induced by pruning. The paper further supports its theoretical claims with extensive experiments on various architectures (MLP, CNN, ViT) and datasets (MNIST, CIFAR10, CIFAR100), offering a principled foundation for designing pruning strategies that not only achieve model compression but also enhance robustness without additional constraints or cost, yielding a “free-lunch” benefit.

**Strengths:**

1. Theoretical novelty and contribution: This paper and provides a relatively comprehensive theoretical analysis of the adversarial risk of pruned DNNs, which is an important and timely topic. The derived bounds and trade-offs provide valuable insights into how pruning affects model robustness, which can guide future research and practical applications.
2. Stable theoretic foundation. This paper provides a clear mathematical formalization of the adversarial risk of DNNs in Equation (3) and detailed proofs in the appendix, which demonstrates the rigor of the theoretical analysis.
3. Extensive empirical validation: The paper supports its theoretical findings with a wide range of experiments across different architectures and datasets. The empirical results align well with the theoretical predictions, strengthening the validity of the claims made in the paper.

**Weaknesses:**

1. Lack of direct experimental results to illustrate the gap between the experimentally measured adversarial risk and the theoretically derived bounds. Specifically, in Theorem 1, the author claims that "the adversarial risk of the pruned model g is always smaller than its corresponding original model f". However, the author only proves, if I don't misunderstand, that the adversarial risk *bound* of g is smaller than that of f. It seems that the latter is not strong enough to support the former.
2. Some figures should be improved; they are too blurry to see clearly. For example, it would be better to enlarge the legend in figure 2.
3. See Question.

**Questions:**

1. Is the definition of adversarial risk of DNNs in Equation (3) widely used or accepted in the AI community?
2. In figure 2, there always exits a "sweet spot" where pruning achieves best attacked accuracy. However, is it always true? Specifically, given a special situation, as pruning ratio increases, if the harm effect of pruning on clean accuracy is larger than the benefit effect of pruning on robustness, then the "sweet spot" may not exist.
3. Can you discuss more about the contributions discussed in Abstract and Conclusion. This paper claims that *"offering a principled foundation for designing pruning strategies that not only achieve model compression but also enhance robustness without additional constraints or cost, yielding a “free-lunch” benefit"*. In fact, this paper seems to not provide any convincing pruning strategies (e.g. how to "chosen pruning parameters") without additional cost, but only analyze existing ones.
4. The models used in this paper are mainly vision models (MLP, ViT, CNN) and vision tasks (MNIST, CIFAR10, CIFAR100). Can the findings be generalized to NLP models and tasks, even LLMs, which are more widely used and need pruning more urgently?

---

> ### Author Response · Authors · 2025-11-20
>
> We are grateful for the reviewer's thoughtful comments and valuable suggestions. We especially appreciate the acknowledgment of our theoretical contributions. We recognize that some aspects of our original submission needed better clarification, which we have addressed in the revision. In what follows, we respond to each point and outline the modifications made.
>
> **Q1 - On the Definition of Adversarial Risk:**  Our definition adapts a classical formulation that has been widely adopted in the literature (e.g., [1, 2], among others). To the best of our knowledge, the core principle of controlling differences in the output manifold to ensure consistent classification under perturbation is well-established in the community. Our adaptation incorporates the probabilistic nature of pruning, which is essential for our theoretical framework and directly captures our goal of improving robustness through pruning.
>
> —
>
> [1] Robustness between the worst and average case. Rice & Al. - NeurIPS 2021
>
> [2] Theoretical evidence for adversarial robustness through randomization. Pinot & Al. NeurIPS 2019.
>
> **W1 - Experimentally validating the Bounds:**  As denoted in the remark after Theorem 1 (Line 239 - 244), we think that comparing the upper bounds obtained through a similar approach is a reasonable way of studying the effect of pruning on robustness in the absence of access to the true theoretical adversarial risk values. Nonetheless, we agree with the reviewer’s assessment that a better analysis of this assumption both through an empirical estimation and the provided theoretical bound is important. In this perspective, we have provided this analysis in Figure 16 (Appendix H.6). Specifically, we first see empirically that by lowering the value of $p$ (increasing the pruning), the empirical adversarial risk diminishes (resulting in better robustness), and also we see that the provided theoretical upper-bound is mimicking a similar behavior as the empirical one, validating therefore the insights in Theorem 1.
>
> **Q3 - A foundation for designing pruning strategies:** We appreciate this observation. Our theoretical analysis connects pruning probabilities to both robustness and model performance through the derived bounds. Building on this foundation, we can formalize the robustness-optimality trade-off as an optimization problem, with weighting terms that allow practitioners to prioritize different objectives based on their use case.
> In the revised manuscript, we have made this framework more concrete. Section 5.3 now provides a clear formalization of this trade-off as a bi-objective optimization problem, while Appendix E details our adapted Coordinate Descent algorithm for solving it to obtain optimal pruning values. We empirically validate both the problem formulation and the algorithm in Section 6.5, demonstrating that the approach successfully identifies optimal pruning configurations that align with the trade-offs observed in our exploratory experiments.
>
> **Q2 - On the clean/attacked sweet spot:**  We appreciate this important question. Building on our theoretical framework, we can indeed use the computed upper bounds and the formulated bi-objective optimization problem to rigorously establish this trade-off. Specifically, the robustness bound is connected to the quantity $p$ while the optimality bound is connected to $1-p$, revealing an inverse relationship between the two objectives.
>
> To formalize this observation, we have derived a theoretical analysis (Appendices F and G) demonstrating the existence of a Pareto front and thus confirming the presence of a sweet spot where both objectives can be balanced. However, we acknowledge the reviewer's important point that such a sweet spot may not always be practically satisfactory. In certain settings, the resulting decrease in clean accuracy may exceed acceptable thresholds for the application at hand. We have incorporated this discussion into Section 5.3 of the revised manuscript to clarify both the theoretical guarantees and their practical limitations.
>
> **Q4 - Extension to LLMs:** We appreciate this suggestion and agree this is a promising direction for future work. We note that our theoretical framework already includes an analysis of Transformer-based models, for which we have derived Lipschitz bounds. Since our framework does not make assumptions about the underlying domain, this provides a foundation for potential extension to LLMs. The primary challenge lies in defining an appropriate distance metric for the discrete nature of text, but once established, similar theoretical insights would follow, enabling us to formalize the trade-off optimization problem and derive optimal pruning strategies.
>
> From an empirical perspective, our approach could be applied to LLMs through systematic search over the pruning parameter space, though computational cost would need to be considered. We believe this extension warrants dedicated investigation and plan to explore it in future work.

---

> > ### Author Response · Authors · 2025-11-28
> >
> > Dear Reviewer,
> >
> > Thank you once again for your valuable time and insightful feedback, which provided valuable guidance for refining our work. We are also particularly grateful for acknowledging the worth our theoretical and empirical insights, and for the positive score. As the discussion period reaches its final part, we would be grateful to know whether our latest responses have resolved your concerns and raised points or if any questions remain. We are happy to provide further clarification as needed.
> >
> > Thanks,
> > Authors

---

### Official Review · Reviewer_gmPM · 2025-10-31

**Soundness:** 1
**Presentation:** 3
**Contribution:** 2
**Rating:** 2
**Confidence:** 4

**Summary:**

This paper provides the theoretical framework explaining how model pruning affects adversarial robustness. It proves that pruning inherently reduces the adversarial risk upper bound, meaning that pruned models can become more robust to small input perturbations. However, pruning also introduces an optimality gap, as excessive sparsity harms clean accuracy.

**Strengths:**

+ Good writing and easy to follow.
+ The theoretical derivation is consistent and well-aligned with the empirical results.
+ The idea of bridging pruning to robustness bounds is interesting and potentially impactful.

**Weaknesses:**

+ Important related work missing. The paper fails to cite a paper that first demonstrated and analyzed the robustness benefits of sparsity [1]. This paper would benefit from a deeper discussion of what new insights are gained beyond [1].
+ Experimental limitations.
    + The experiments use too few attack methods (only PGD/FGSM). More recent and diverse attacks (AutoAttack, CW, etc.) should be included to make the conclusions generalizable.
    + The evaluation benchmarks and models are relatively small (e.g., MLPs, small CNNs). Results on larger and more modern architectures would make the paper more convincing.
+ I'm just wondering whether activation pruning/sparsity will have similar phenomenon?

[1] Guo, Yiwen, et al. "Sparse dnns with improved adversarial robustness." Advances in neural information processing systems 31 (2018).

**Questions:**

Please see weaknesses.

---

> ### Author Response · Authors · 2025-11-20
>
> We thank the reviewer for their constructive feedback, which has helped strengthen our manuscript. We particularly appreciate the recognition of our theoretical contributions and the helpful identification of areas requiring clarification. Below, we address the raised concerns and outline the revisions made.
>
> **W1 - Regarding previous work:** We appreciate the reviewer pointing us to this highly relevant reference. We agree that our work extends this line of research in meaningful ways. While the previous work [1] demonstrated that sparsity can incidentally improve adversarial robustness (theoretically) and empirically observed the clean/attacked accuracy trade-off, our work advances this by making pruning itself the focus of theoretical analysis. Specifically, we contribute:
> - We establish formal relationships between pruning parameters and both robustness and optimality (reflecting clean accuracy);
> - We formulate the trade-off as a bi-objective optimization problem (now better presented in Section 5.3) and provide an adapted Coordinate Descent algorithm to solve it (detailed in Appendix E, validated in Section 6.5);
> - We rigorously analyze the existence of a "sweet spot" with favorable Pareto trade-offs (Appendices F and G), providing theoretical grounding for the empirical observations in prior work.
>
> Following the reviewer's suggestion, we have added a dedicated discussion in Section 7 clarifying how our work builds upon [1]. We introduce a theoretical lens on common pruning rules and derive Lipschitz-style bounds showing that, under shared assumptions, pruning monotonically reduces adversarial risk for MLPs and Transformers. We further frame pruning as a bi-objective optimization problem over layerwise sparsity, allowing systematic exploration of the robustness-accuracy trade-off. Our empirical results across multiple architectures and modalities support the theory and reveal how architectural choices shape the sweet spot where pruning improves robustness without degrading accuracy.
>
>
> **W2 - Regarding the Experimental Setting:** We appreciate the reviewer's suggestions for strengthening our empirical validation. We would like to clarify that our original manuscript included results on Vision Transformers (ViT), which we consider a substantial architecture for demonstrating generalizability or larger architectures.
> Regarding attack diversity, we agree this is an important consideration. While our original manuscript focused on FGSM and PGD to illustrate the existence of the trade-off, we have now extended our evaluation to include the CW Attack and AutoAttack. These additional experiments, included in the revised manuscript (Appendix H.5), demonstrate that our findings generalize across a broader range of adversarial perturbations.
>
> **W3 - On the Effect of Activation Pruning/sparsity:** We appreciate this interesting observation. Our theoretical framework assumes classical activation functions that are 1-Lipschitz. Activation functions that induce sparsity (such as ReLU variants with thresholding) would likely yield tighter Lipschitz bounds, potentially enhancing the robustness gains predicted by our analysis. However, the key challenge with aggressive activation sparsity lies in maintaining sufficient expressive power to preserve model performance. As elaborated in the revisited manuscript, our framework suggests that any pruning mechanism must navigate the fundamental trade-off between robustness and accuracy. Exploring activation-level sparsity could be a valuable direction for future work.

---

> > ### Author Response · Authors · 2025-11-28
> >
> > Dear Reviewer,
> >
> > Thank you once again for your valuable time and insightful feedback, which provided valuable guidance for refining our work. As the discussion period reaches its final part, we would be grateful to know whether our latest responses have resolved your concerns and raised points or if any questions remain. We are happy to provide further clarification as needed.
> >
> > Thanks,
> > Authors

---

### Meta-Review · Area_Chair_HzGb · 2026-01-07

**Summary:**

This paper studies the relationship between model pruning and adversarial robustness. It provides a theoretical analysis linking pruning probabilities to upper bounds on adversarial risk and to deviation from the original model’s performance, and frames the resulting robustness–accuracy trade-off as a bi-objective optimization problem.
Several reviewers found the mathematical analysis sound and appreciated the attempt to move beyond purely empirical observations. However, the reviews raised substantial concerns about the paper’s positioning relative to prior work, the limited robustness evaluation (FGSM/PGD only), unclear practical implications of the theory, and whether the results offered genuinely new insight beyond the well-known robustness–accuracy trade-off.

**Reviewer Concerns:**

### Concerns addressed by the rebuttal:

- The authors clarified that their robustness definitions and added discussion to better position the contribution relative to prior work on pruning and robustness.

- Experiments were extended to CW and AutoAttack, which is more promising.

- Additional analysis was added comparing empirical adversarial risk to the theoretical upper bounds, helping justify the relevance of the theory.

- Concerns about uniform pruning were addressed through discussion and experiments on layer-wise pruning, along with clarification of how model redundancy affects observed accuracy.


### Concerns that remain outstanding:

- Not clear whether the theoretical insights proposed can be impactful beyond existing empirical methods.

- The scope is limited to post-training, unstructured pruning, which is hard to really decrease the computational cost.

- The theoretical treatment does not fully extend to architectures with more complex structures.

**Reviewer Scores:**

Reviewer gmPM: Initially 2. Given the added experiments and clarifications, this reviewer might increase score to 4.

Reviewer fjkr: Initially 2. After rebuttal, explicitly raised the score to 4, and indicated openness to further change depending on overall consensus.

Reviewer w3kp: Initially 6. Concerns were partially addressed; score likely unchanged.

Reviewer BmZv: Initially 6. Additional experiments and clarifications address most concerns; score likely unchanged.

Reviewer 9Euk: Initially 6. Follow-up concerns were resolved during discussion; score remains unchanged.

---

### Decision · Program_Chairs · 2026-01-26

Reject